# On Convergence of Polynomial Approximations to the Gaussian Mixture Entropy

**Caleb Dahlke**
Department of Mathematics
University of Arizona

**Jason Pacheco**
Department of Computer Science
University of Arizona

## Abstract

Gaussian mixture models (GMMs) are fundamental to machine learning due to their flexibility as approximating densities. However, uncertainty quantification of GMMs remains a challenge as differential entropy lacks a closed form. This paper explores polynomial approximations, specifically Taylor and Legendre, to the GMM entropy from a theoretical and practical perspective. We provide new analysis of a widely used approach due to Huber et al. (2008) and show that the series diverges under simple conditions. Motivated by this divergence we provide a novel Taylor series that is provably convergent to the true entropy of any GMM. We demonstrate a method for selecting a center such that the series converges from below, providing a lower bound on GMM entropy. Furthermore, we demonstrate that orthogonal polynomial series result in more accurate polynomial approximations. Experimental validation supports our theoretical results while showing that our method is comparable in computation to Huber et al. We also show that in application, the use of these polynomial approximations, such as in Nonparametric Variational Inference, rely on the convergence of the methods in computing accurate approximations. This work contributes useful analysis to existing methods while introducing novel approximations supported by firm theoretical guarantees.

## 1 Introduction

Entropy is a natural measure of uncertainty and is fundamental to many information-theoretic quantities such as mutual information (MI) and Kullback-Leibler (KL) divergence [8]. As a result, entropy plays a key role in many problems of ML including model interpretation [7], feature selection [6], and representation learning [27]. It is often used in the data acquisition process as in active learning [25, 26], Bayesian optimal experimental design [17, 5, 3], and Bayesian optimization [13]. Yet, despite its important role entropy is difficult to calculate in general.

One such case is the Gaussian mixture model (GMM), where entropy lacks a closed form and is the focus of this paper. GMMs are fundamental to machine learning and statistics due to their property as universal density approximators [18]. However, the lack of a closed-form entropy requires approximation, often via Monte Carlo expectation. Such stochastic estimates can be undesirable as computation becomes coupled with sample size and a deterministic approach is often preferred. Simple deterministic bounds can be calculated via Jensen's inequality or Gaussian moment matching [14]. Such bounds are often too loose to be useful, leading to other options such as variational approximations [21, 10] and neural network-based approximation [2]. Yet, these deterministic estimators do not allow a straightforward tradeoff of computation and accuracy as in the Monte Carlo setting.

Polynomial series approximations are both deterministic and provide a mechanism for computation-accuracy tradeoff by varying the polynomial degree. In this paper we focus on three such polynomial approximations of the GMM entropy. We begin with the widely used approximation of Huber et al. (2008). While this approximation yields good empirical accuracy in many settings, a proof of

37th Conference on Neural Information Processing Systems (NeurIPS 2023).

convergence is lacking. In this work we show that the Huber et al. approximation in fact does not converge in general, and we provide a divergence criterion (Theorem 3.1). In response to the divergent behavior, we propose two alternative polynomial approximations, a Taylor and Legendre series approximation of GMM entropy that are provably convergent. We establish in Theorem 4.2 and Theorem 4.5 that each series converges everywhere under conditions on the center point or interval, respectively. In Theorem 4.4 we provide a simple mechanism for choosing a value to ensure that these series converge everywhere. We additionally establish, in Theorem 4.3, that our Taylor approximation is a convergent lower bound on the true entropy for any finite poynomial order.

The complexity of both Huber et al. and our proposed methods have similar computation largely driven by polynomial order. To address this we propose an approximation that estimates the higher-order terms by fitting a polynomial regression. This approach requires the evaluation of only three consecutive polynomial orders to approximate higher order series. In this way we can obtain more accurate estimates without the computational overhead of evaluating higher order polynomial terms.

We conclude with an empirical comparison of all polynomial approximations that produce the divergent behavior of the Huber et al. approximation while our propsed methods maintain convergence. We also compare accuracy and compuation time for each method accross a varaitey of dimensions, number of GMM components, and polynomial orders. Finally, we show an application of our methods in Nonparametric Variational Inference [11] where the guarantees of convergence play a large role in the accuracy of posterior approximation via GMMs.

## 2   Preliminaries

We briefly introduce required notation and concepts, beginning with a definition of the Gaussian mixture entropy. We will highlight the challenges that preclude efficient computation of entropy. We conclude by defining notation that will be used for discussion of polynomial approximations.

### 2.1   Gaussian Mixture Entropy

The differential entropy of a continuous-valued random vector $x \in \mathbb{R}^d$ with a probability density function $p(x)$ is given by,

$$H(p(x)) = - \int p(x) \log p(x) dx = \mathbb{E}[-\log p(x)]. \tag{1}$$

The differential entropy is in $[-\infty, \infty]$ for continuous random variables. It is a measure of uncertainty in the random variable in the sense that its minimum is achieved when there is no uncertainty in the random vector, i.e., a Dirac delta, and approaches the maximum as the density becomes uniformly distributed.

Gaussian mixtures are ubiquitous in statistics and machine learning due to their property as universal density approximators [18]. However, despite this flexibility, the entropy of a Gaussian mixture requires computing the expectation of the log-sum operator, which lacks a closed form. Many approximations and bounds are used in practice. A simple upper bound is given by the entropy of a single Gaussian with the same mean and covariance as the mixture [14], and a lower bound can be obtained by Jensen's inequality. Though efficient, these bounds are very loose in practice, leading to more accurate methods being used, such as various Monte Carlo approximations, deterministic sampling [12], and numerous variational bounds and approximations [21, 10].

### 2.2   Taylor Polynomials

In this paper we explore entropy approximation using Taylor polynomials. The $n^{th}$-order Taylor polynomial of a function $f(x)$ with evaluation point $c$ is given by,

$$T_{f,n,c}(x) = \sum_{i=0}^{n} \frac{f^{(n)}(c)}{n!}(x - c)^n, \tag{2}$$

where $f^{(n)}(c)$ denotes the $n^{\text{th}}$ derivative of $f$ evaluated at point $c$. The Taylor series has a region of convergence which determines the range of x-values where the series accurately represents the original function. It depends on the behavior of the function and its derivatives at the expansion point. Analyzing the region of convergence is crucial for ensuring the validity of the Taylor series approximation. Various convergence tests, such as the ratio test, help determine the x-values where the Taylor series provides an accurate approximation.

## 2.3 Orthogonal Polynomials

Taylor series are versatile approximations, however predominately behave well near the center point chosen. We ideally would like an approximation that performs well across a range of values. To achieve this, we consider series approximation via orthogonal polynomials. A set of orthogonal polynomials on the range $[a, b]$ is an infinite sequence of polynomials $P_0(x), P_1(x), \ldots$ where $P_n(x)$ is an $n^{th}$ degree polynomial and for any pair of polynomials satisfies

$$\langle P_i(x), P_j(x) \rangle = \int_a^b P_i(x) P_j(x) dx = c_i \delta_{ij} \tag{3}$$

where $\delta_{ij}$ is the Kronecker delta function and $c_i$ is some constant. Orthogonal polynomials can be used to approximate a function, $f(x)$, on their interval, $[a, b]$, by finding the projection of $f(x)$ onto each polynomial in the series $P_i(x)$.

$$f(x) = \sum_{i=1}^{\infty} \frac{\langle f(x), P_i(x) \rangle}{\langle P_i(x), P_i(x) \rangle} P_i(x) \tag{4}$$

Any appropriate choice of orthogonal polynomials can be used. One might be interested in considering the Chebyshev polynomials for their property of minimizing interpolation error or Legendre polynomials for their versatility and ease of computation.

# 3 Convergence of Polynomial Approximations

To estimate the entropy $H(p) = \mathbb{E}_p[-\log(p(x))]$ using a polynomial approximation one may approximate either the log-density $\log(p(x))$ or just the logarithm $\log(y)$. We will show that estimating $\log(p(x))$ has convergence issues and that it can be complicated to compute due to tensor arithmetic in higher dimensions. Both of these issues will be addressed by simply approximating $\log(y)$ and computing the exact $p(x)$. All proofs are deferred to the Appendix for space.

## 3.1 Divergence of Huber et al. Approximation

We begin our exploration with a widely used approximation of the GMM entropy due to [15]. Let $p(x)$ be a GMM and the log-GMM $h(x) = \log(p(x))$. Huber et al. provides a Taylor series approximation of the GMM entropy given by,

$$\log(p(x)) = -\sum_{i=1}^{M} w_i \sum_{n=0}^{\infty} \frac{h^{(n)}(\mu_i)}{n!} (x - \mu_i)^n, \tag{5}$$

The series is $M$ individual Taylor series evaluated at each component mean, $\mu_i$. The equality in Eqn. (5) only holds if the series converges, which we will show is not the case in general.

**Theorem 3.1** (Divergence Criterion for Huber et al.). *Let $p(x) = \sum_{i=1}^{M} w_i \mathcal{N}(x \mid \mu_i, \Sigma_i)$ and consider the Taylor series presented in Eqn. (5). If any mean component, $\mu_i$, satisfies the condition $p(\mu_i) < \frac{1}{2} max(p(x))$, then the series diverges, otherwise it converges.*

Theorem 3.1 provides us with the condition that Huber et al.'s approximation Eqn. (5) will diverge. This means that the entropy approximation will be inaccurate for any GMM with any of its modes less than half the probability of any other point, as illustrated in Fig. 1.

## 3.2 Taylor Series Approximation of the Logarithm

Motivated by the divergence of the previous Taylor series we propose a different approach that is provably convergent. While Huber et al. perform a Taylor decomposition of the log-GMM PDF, our approach decomposes only the $\log(y)$ function using a Taylor series of the function centered about the point $a$. It is well-known that this series converges for values $|y - a| < a$ [28] and is given by,

$$\log(y) = \log(a) + \sum_{n=1}^{\infty} \frac{(-1)^n}{na^n} (y - a)^n. \tag{6}$$

Note the change of $c$ to $a$ as the Taylor series center. This change highlights the difference in function domains. In particular, the former series is computed on values of the random vector $x$, whereas ours is computed on the PDF $y = p(x)$. Choosing any center $a > \frac{1}{2} max(p(x))$ will ensure that the series converges everywhere.

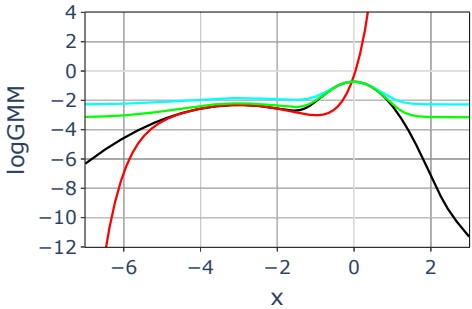 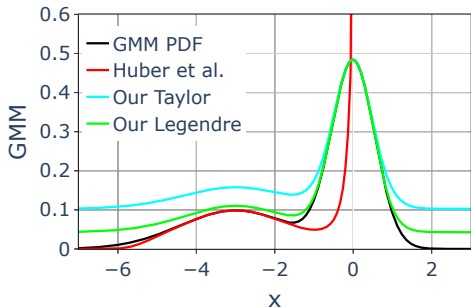

Figure 1: **Divergence of Huber et al.** and convergence of our polynomial series approximation are plotted for the Gaussian mixture, $p(x) = .35\mathcal{N}(x \mid -3, 2) + .65\mathcal{N}(x \mid 0, .2)$. In the left graph, the log-GMM is plotted, which is what each series is defined for. The right plot is the exponential of the series so we can see how each converge in the more familiar framework of a GMM. Notice that the Huber et al. is centered on the first component mean $\mu_1 = -3$ and diverges around the mean of the second component $\mu_2 = 0$ as supported by Theorem 3.1 since the mode $\mu_1$ is less than half the probability at the mode $\mu_2$. Both of our methods are convergent, the Taylor series is a bound (Theorem 4.3) while the Legendre series has a lower global error.

**Lemma 3.2** (Convergent Taylor Series of Log). *If $a > \frac{1}{2}max(p(x))$, then for all $x$*

$$\log(p(x)) = \log(a) + \sum_{n=1}^{\infty} \frac{(-1)^{n-1}}{na^n} \sum_{k=0}^{n} \binom{n}{k}(-a)^{n-k}p(x)^k \tag{7}$$

The proof of Lemma 3.2 is a simple ratio test. The only assumption on $p(x)$ is that it has a finite maximum, which is true for any non-degenerate GMM with positive definite component covariances. As a result, the Taylor series converges for all $x$ regardless of the GMM form.

### 3.3 Legendre Series Approximation of the Logarithm

For the orthogonal polynomial approximation, we consider the Legendre polynomials, specifically the shifted Legendre polynomials [4] which are orthogonal on $[0, a]$,

$$P_n(y) = L_{[0,a],n}(y) = \sum_{k=0}^{n} \frac{(-1)^{n+k}(n+k)!}{(n-k)!(k!)^2 a^k} y^k \tag{8}$$

**Lemma 3.3** (Convergent Legendre Series of Log). *If $a > max(p(x))$, and consider the shifted Legendre polynomials on the interval $[0, a]$ in Eqn. (8). Then for $x$ a.e.*

$$\log(p(x)) = \sum_{n=0}^{\infty}(2n+1) \sum_{j=0}^{n} \frac{(-1)^{n+j}(n+j)!((j+1)\log(a) - 1)}{(n-j)!((j+1)!)^2} L_{[0,a],n}(p(x)) \tag{9}$$

Again, all that is assumed about this approximation is that the max of a GMM can be bounded, so this approximation converges for all GMMs regardless of structure.

## 4 GMM Entropy Approximations

Having established multiple polynomial approximations in Sec. 3, we now consider applying them to the definition of entropy for a GMM. We can directly substitute the series approximation into the entropy definition, $H(p(x)) = \mathbb{E}_p[-\log(p(x))]$, and push the expectation through the summations.

### 4.1 Huber et al. Entropy approximation

Applying Huber et al.'s Taylor series approximation of the $\log(p(x))$, we see the GMM entropy can be approximated by,

$$H(p(x)) = -\sum_{i=1}^{m} w_i \sum_{n=0}^{\infty} \frac{h^{(n)}(\mu_i)}{n!} \mathbb{E}_{q_i}[(x - \mu_i)^n], \tag{10}$$

where $q_i(x) = \mathcal{N}(x \mid \mu_i, \Sigma_i)$ is shorthand for the $i^{\text{th}}$ Gaussian component. The attractive feature of Eqn. (10) is that it simplifies the expected value of a log-GMM to the $n^{\text{th}}$ central moments of the $i^{\text{th}}$

component which, is exactly zero when $n$ is odd and has a closed form when $n$ is even. However, Theorem 3.1 shows that this approximation is not guaranteed to converge, which is supported by experimental results in Sec. 6. Furthermore, in higher dimensions, $h^{(n)}(\mu_i) = \frac{\partial^n h(\mu_i)}{\partial x_1^{j_1} \dots \partial x_d^{j_n}}$, where $j_1 + \dots + j_d = n$ which grows rapidly, is an $n$ dimensional tensor. This is cumbersome to compute and is difficult to deal with the tensor arithmetic required beyond a Hessian. In practice, this limits Eqn. (10) to only second order approximations for random vectors.

## 4.2 Taylor Series Entropy Approximation

Having established the convergent Taylor series of the logarithm in Lemma 3.2, we can apply the approximation and push the expectation through the summations. This reduces the computation of the entropy to computing $\mathbb{E}_p[p(x)^k]$ for all $k < n$ where $n$ is the order of the polynomial approximation.

**Lemma 4.1** (Closed form expectation of powers of GMMs). *Let $p(x) = \sum_{i=1}^M w_i \mathcal{N}(x|\mu_i, \Sigma_i)$ be a GMM and $k$ be a non-negative integer. Then*

$$\mathbb{E}_p[p(x)^k] = \sum_{j_1 + \dots + j_M = k} \binom{k}{j_1, \dots, j_M} \sum_{i=1}^M w_i \left( \frac{\mathcal{N}(0|\mu_i, \Sigma_i)}{\mathcal{N}(0|\mu, \Sigma)} \prod_{t=1}^M (w_t \mathcal{N}(0|\mu_t, \Sigma_t)^{j_t}) \right) \quad (11)$$

*where $\Sigma = (\Sigma_i^{-1} + \sum_{t=1}^M j_t \Sigma_t^{-1})^{-1}$ and $\mu = \Sigma(\Sigma_i^{-1}\mu_i + \sum_{t=1}^M j_t \Sigma_t^{-1}\mu_t)$.*

While Eqn. (11) may seem complicated at first glance, it is straightforward to compute. All terms are Gaussian densities, polynomial functions, and binomial coefficients. Lemma 4.1 is defined for $\mathbb{E}_p[p(x)^k]$ but an analogous definition holds for $\mathbb{E}_p[q(x)^k]$ allowing us to apply all the following results not only to entropy, but cross-entropy, KL and MI of GMMs. This is not the focus of this paper, however a discussion can be found in A.4 for completeness. Using Lemma 3.2 and Eqn. (11), we can obtain the following approximation,

$$\hat{H}_{N,a}^T(p(x)) = -\log(a) - \sum_{n=1}^N \frac{(-1)^{n-1}}{na^n} \sum_{k=0}^n \binom{n}{k}(-a)^{n-k}\mathbb{E}_p[p(x)^k] \quad (12)$$

To ensure the expected value can be pushed through the infinite sum of the series, we check that our finite order entropy approximation does still converge to the true entropy.

**Theorem 4.2** (Convergence of $\hat{H}_{N,a}^T(p(x))$). *Let $p(x) = \sum_{i=1}^M w_i \mathcal{N}(x|\mu_i, \Sigma_i)$ be a GMM and choose a Taylor center such that $a > \frac{1}{2}max(p(x))$. Then, for $\hat{H}_{N,a}^T(p(x))$ defined in Eqn. (12)*

$$\lim_{N \to \infty} \hat{H}_{N,a}^T(p(x)) = H(p(x)) \quad (13)$$

Having established convergence of our estimator, it remains to provide a method for selecting a Taylor center that meets the convergence criterion $a > \frac{1}{2}max(p(x))$. In fact, we show in Theorem 4.3 that selecting a looser condition $a > max(p(x))$ ensures convergence from below, thus yielding a lower bound on the true entropy.

**Theorem 4.3** (Taylor Series is Lower Bound of Entropy). *Let $p(x) = \sum_{i=1}^M w_i \mathcal{N}(x|\mu_i, \Sigma_i)$ and $a > max(p(x))$. Then, for all finite $N$,*

$$\hat{H}_{N,a}^T(p(x)) \leq H(p(x)) \quad (14)$$

We have now established that the Taylor center chosen as $a > max(p(x))$ is both convergent and yields a lower bound. In fact, it is easy to find such a point by upper bounding the maximum of a GMM as given in Theorem 4.4.

**Theorem 4.4** (Upper bound on maximum of a GMM). *Let $p(x) = \sum_{i=1}^M w_i \mathcal{N}(x|\mu_i, \Sigma_i)$, then*

$$max(p(x)) \leq a = \sum_i^M w_i |2\pi\Sigma_i|^{-\frac{1}{2}} \quad (15)$$

In our experience choosing a center closer to the convergence criterion $a > \frac{1}{2}max(p(x))$ yields slightly more accurate estimates, but not significantly so.

### 4.3 Legendre Entropy Approximation

Now, starting with the convergent Legendre approximation considered in Lemma 3.3 and Eqn. (11), we can obtain the following approximation,

$$\hat{H}_{N,a}^L(p(x)) = -\sum_{n=0}^{N}(2n+1)\sum_{j=0}^{n}\frac{(-1)^{n+j}(n+j)!((j+1)\log(a)-1)}{(n-j)!((j+1)!)^2}L_{[0,a],n}\left(\mathbb{E}_p[p(x)^k]\right) \quad (16)$$

Again, we verify that taking the expectation of our series does not effect convergence.

**Theorem 4.5** (Convergence of $\hat{H}_{N,a}^L(p(x))$). *Let $p(x) = \sum_{i=1}^{M}w_i\mathcal{N}(x|\mu_i,\Sigma_i)$ be a GMM and choose an interval such that $a > max(p(x))$. Then for $\hat{H}_{N,a}^L(p(x))$ defined in Eqn. (16)*

$$\lim_{N\to\infty}\hat{H}_{N,a}^L(p(x)) = H(p(x)) \quad (17)$$

Now having established converge criterion for the Legendre series approximation, we need to choose an upper point of the interval for the Legendre series. We need to choose $a > max(p(x))$ which is satisfied by the same $a$ found in Lemma 4.4.

### 4.4 Approximation of Taylor Series Limit

Computing the series in Eqn. (12) for higher orders can be computationally prohibitive. In particular, the sum $\sum_{j_1+...+j_M=n}$ is over $M$ integers summing to $n$, which is $\mathcal{O}((n+M-1)!)$. In this section, we provide an approximation that avoids explicit computation of this sum for higher orders. We employ a method (similar to Richardson extrapolation [22]) that is based on a polynomial fit of the convergence rate for the lower bound property discussed in Theorem 4.3. From Taylor's theorem, we know that a function can be represented as $f = T_n + R_n$, where $T_n$ is the $n^{th}$ order Taylor polynomial and $R_n = \mathcal{O}(\alpha^n)$ is the remainder. Rewriting this in terms of the Taylor polynomial, we observe that $T_n = -R_n + f = \beta\alpha^n + \eta$. We take $\beta < 0$ to represent the negative scale factor in front of the remainder, and $0 < \alpha < 1$ to model the decay of the remainder.

$$\hat{H}_{n,a}^T(p(x)) = \beta \cdot \alpha^n + \eta \quad (18)$$

We require three consecutive orders of our Taylor series approximation, $\hat{H}_{n,a}^T(p(x))$, $\hat{H}_{n+1,a}^T(p(x))$, and $\hat{H}_{n+2,a}^T(p(x))$, to solve for the three unknown parameters:

$$\hat{H}_{n,a}^T(p(x)) = \beta \cdot \alpha^n + \eta$$
$$\hat{H}_{n+1,a}^T(p(x)) = \beta \cdot \alpha^{n+1} + \eta$$
$$\hat{H}_{n+2,a}^T(p(x)) = \beta \cdot \alpha^{n+2} + \eta.$$

Since $0 < \alpha < 1$, $\lim_{n\to\infty}\beta\alpha^n = 0$, indicating that we aim to solve for $\eta$ as our approximation of the limit of the Taylor series entropy.

$$\hat{H}_{N,a}^{TL}(p(x)) = \eta = \hat{H}_{N-2,a}^T(p(x)) - \frac{(\hat{H}_{N-1,a}^T(p(x)) - \hat{H}_{N-2,a}^T(p(x)))^2}{\hat{H}_{N,a}^T(p(x)) - 2\hat{H}_{N-1,a}^T(p(x)) + \hat{H}_{N-2,a}^T(p(x))} \quad (19)$$

This approach assumes that the Taylor series converges according to Eqn. (18), which is not the case in general. Identifying the exact rate of convergence is a topic of future work. Nevertheless, this simple approximation has shown higher accuracy in practice with negligible additional computation, as demonstrated in the experiments of Sec. 6. With a slight abuse of terminology, we refer to this approach as the *Taylor limit*. We do not apply this method to the Legendre approximation (Eqn. (16)) as it doesn't maintain a lower bound during its convergence. Although equivalent methods have been considered to model potential oscillation convergence, in practice, we do not find an increase in accuracy.

## 5 Related Work

Numerous approximation methods exist in the literature for estimating entropy and related information measures, such as mutual information and Kullback-Leibler divergence, in the context of Gaussian Mixture Models (GMMs). Monte Carlo estimation, deterministic bounds using Jensen's inequality,

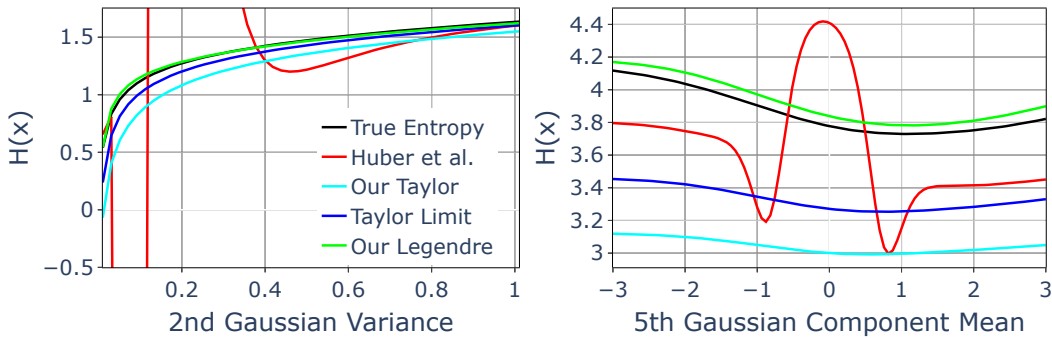

Figure 2: **Scalar GMM** example is plotted on the left. The variance of a component of a two component GMM is varied between $\sigma_2^2 \in (0, 1]$ as in theory according to Theorem 3.1, the example will be divergent where $\sigma_2^2 < .46$ and convergent above. We plot the fourth order of each method and see that Huber et al.'s approximation does diverge where the theory predicts. **Two dimensional GMM** with five components is consider on the right. Here the mean of a single component is shifted from $\mu_5 = [-3, -3]^T$ to $\mu_5 = [3, 3]^T$. We consider the third order approximation of each method and see that Huber et al. is poorly behaved. In both examples, we see that our Taylor method is a lower bound (Theorem 4.3), the Taylor limit provides higher accuracy, and Our Legendre method is a highly accurate approximate.

best-fit moment matched Gaussians, and numerical integral approximations based on the unscented transform have been explored [16, 14, 15]. This paper focuses on the Taylor approximation by Huber et al., an alternative Taylor approximation is proposed by Sebastiani [24], which assumes a shared covariance matrix among GMM components in the high variance setting. However, neither Huber et al. or Sebastiani provide theoretical analysis or convergence guarantees offered in our present work. An analysis conducted by Ru et al. [23] explores the efficiency of Huber et al.'s method and demonstrates that deterministic quadrature methods can be equally fast and accurate in a single dimension, however quadrature methods scale poorly with dimension, at $\mathcal{O}(N^D)$ where $N$ is the number of quadrature points per dimension and $D$ is the dimension of the problem.

Variational approximations and bounds are also widely explored for estimating entropy and mutual information (MI). Much of this work is motivated by the use of Gibbs' inequality, which leads to bounds on entropy and MI [1]. Later work explored similar techniques for upper and lower bounds on MI [21, 10]. More recent work uses artificial neural networks (ANNs) as function approximators for a variety of information-theoretic measures based on differential entropy. The MI neural estimator (MINE) uses such an approach for representation learning via the *information bottleneck* [2] based on the Donsker-Varadhan (DV) lower bound on KL [9]. Related methods use ANNs for optimizing the convex conjugate representation of Nguyen et al. [20]. McAllester and Stratos [19] show that many of these distribution-free approaches based on ANN approximation rely on Monte Carlo approximations that have poor bias-variance characteristics which they provide their own Difference of Entropies (DoE) estimator that achieves the theoretical limit on estimator confidence.

## 6 Experiments

We consider two experiments, a synthetic GMM section where we look at divergence of Huber et al. approximation (Eqn. (10)) and convergence of our three methods, our Taylor (Eqn. (12)), Taylor limit (Eqn. (19)), and our Legendre (Eqn. (16)). Furthermore, we give comparisons of accuracy and computation time across a variety of setting of approximation order, number of GMM components, and dimension for all methods. We then show our how our methods can be applied in practice to Nonparametric Variational Inference [11] where the convergence guarantees of the estimators has a noticeable accuracy improvement on their algorithm.

### 6.1 Synthetic Multivariate GMM

To highlight the theoretical properties, such as convergence, divergence, accuracy, and lower-bound of methods as discussed in Sec. 4, we will consider some synthetic GMMs. We create two GMMs similar to the example published in [15] (original experiment recreated in A.5). We consider a single and multi-dimensional dimensional case that satisfy the divergence criterion in Theorem 3.1. We also look at a time and accuracy analysis versus dimension, components, and polynomial order.

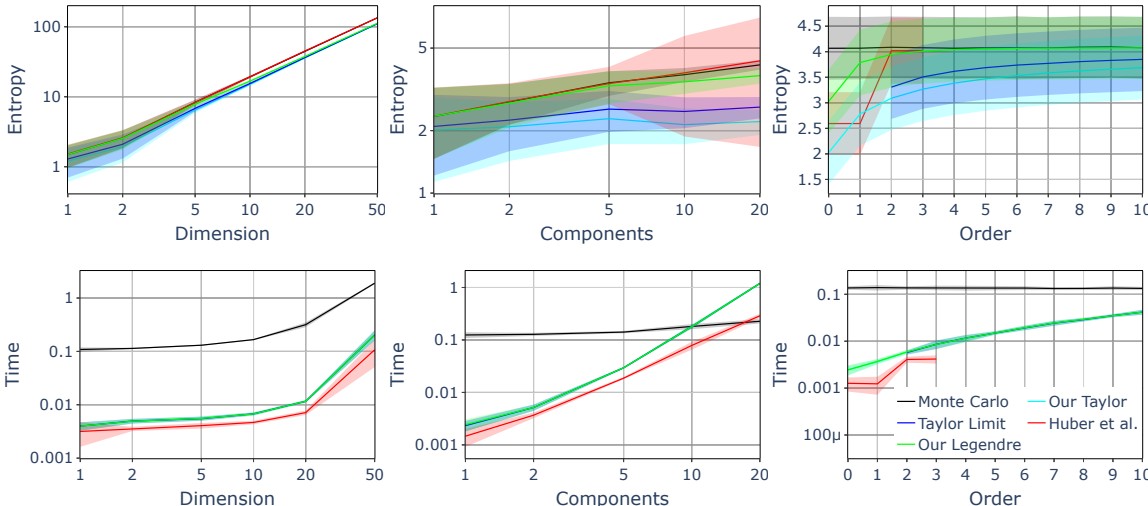

Figure 3: **Dimension (left)** of a two-component GMM varies from zero to fifty, for the second order of each method. Our methods show comparable accuracy and computation time to Huber, regardless of dimension. **Number of components (middle)** in a two-dimensional GMM is considered for the second order approximation of all methods. Huber et al. and our Legendre approximations are nearly equivalent in accuracy, while the Taylor series and Taylor limit serve as lower bounds. Computation time for all our methods is identical and comparable to Huber et al., deviating only at high numbers of components. **Order (right)** of each approximation is varied for a three-dimensional, two-component GMM. Huber et al. is plotted up to order three, as higher orders are restrictive due to tensor arithmetic and Taylor limit starts at order two as it requires three consecutive terms.

**Scalar GMM** In this experiment, we consider a scalar GMM as fourth order and above cannot be easily computed in higher dimensions for Huber et al. due to tensor arithmetic. We use a simple two-component GMM with parameters $w_1 = 0.35$, $w_2 = 0.65$, $\mu_1 = -2$, $\mu_2 = -1$, $\sigma_1^2 = 2$, and $\sigma_2^2 \in (0, 1]$. We are changing the variance of the second Gaussian, $\sigma_2^2$, in the range $(0, 1]$ because the condition for divergence in Theorem 3.1 ($p(x = \mu_1) < \frac{1}{2}p(x = \mu_2)$) is satisfied approximately when $\sigma_2^2 < 0.46$ meaning this experiment should have regions of both convergence and divergence for Huber et al. approximation. Fig. 2 (left) shows the fourth order approximations of all methods. We see that the Huber et al. approximations diverges as expected in the range where $\sigma_2^2 < .46$. Our Taylor method remains convergent and accurate for all values while maintaining a lower bound. Again, our limit method gains us some accuracy and still manages to be lower bound. In this case, we see that the Legendre approximation is a near perfect fit for the entropy.

**Multivariate GMM** To demonstrate that divergence is not limited to single dimension or higher orders, we consider a five-component, two-dimensional GMM with the parameters $w_i = .2\ \forall i$, $\mu_1 = [0, 0]^T$, $\mu_2 = [3, 2]^T$, $\mu_3 = [1, -.5]^T$, $\mu_4 = [2.5, 1.5]^T$, $\mu_5 = c[1, 1]^T$ for $c \in [-3, 3]$, $\Sigma_1 = .25\mathbf{I}_2$, $\Sigma_2 = 3\mathbf{I}_2$, and $\Sigma_3 = \Sigma_4 = \Sigma_5 = 2\mathbf{I}_2$ where $\mathbf{I}_2$ is the two dimensional identity matrix. This examples shifts the mean of the fifth component to show that simply the location of components can make the Huber et al. approximation behave poorly. Fig. 2 (right) shows the third order approximation of each method. We see that Huber et al. is clearly not well behaved in this case even with low order approximation. Furthermore, we continue to see a lower bound by our Taylor method, an increased accuracy from out limit method, and that the Legendre approximation is very close to the true entropy.

### 6.1.1 Computation Time

In this experiment we empirically analyze the computation time of each method as a function of Gaussian dimension, number of Gaussian components, and the order of each polynomial approximation. The baseline of each method will be compared to the Monte Carlo estimation of entropy using $L = 1000$ samples $\{x_j\}_{j=1}^{L} \sim p$. The Monte Carlo estimator is given by $\hat{H} = \frac{1}{L}\sum_j(-\log p(x_j))$.

**Dimension** In Fig. 3 (left), we evaluate the accuracy and computation time for 30 two-component GMMs per dimension in the range of $[1, 50]$. Comparing second order approximations of all methods against the Monte Carlo estimator, our polynomial approximations demonstrate similar accuracy and

nearly identical computation time. The results are comparable to Huber, indicating that our methods preserve accuracy and computation efficiency while providing convergence guarantees.

**GMM Components**  In Fig. 3 (middle), accuracy and computation time are presented for 30 two-dimensional GMMs with varying numbers of components (from 1 to 20) using second order approximations. Legendre and Huber methods show slightly higher accuracy compared to our Taylor approximation and Taylor limit. Notice, Huber's standard deviation also increases with more components, due to the increased likelihood of satisfying the divergence condition in Theorem 3.1. Computation time remains similar for all methods, but is more prohibitive for higher components.

**Polynomial Order**  Fig. 3 (right) shows as the order of the polynomial approximation increases for two-component GMMs in three dimensions. Legendre and Huber methods show higher accuracy compared to Taylor approximation and Taylor limit. Huber is limited to order 3 due to Tensor arithmetic, while Taylor limit starts at order 2 as it requires multiple orders. Computation times are similar across all methods. Notice no accuracy is gained from zero to first order and from second to third order in Huber's approximation due to relying on odd moments of Gaussians which are zero.

## 6.2   Nonparametric Variational Inference

Consider a target density $p(x, \mathcal{D})$ with latent variables $x$ and observations $\mathcal{D}$. The NPV approach [11] optimizes the evidence lower bound (ELBO), $\log p(x, \mathcal{D}) \geq \max_q H_q(p(x, \mathcal{D})) - H_q(q(x)) \equiv \mathcal{L}(q)$ w.r.t. an $m$-component GMM variational distribution $q(x) = \frac{1}{N} \sum_{i=1}^{m} \mathcal{N}(x|\mu_i, \sigma_i^2 I_d)$. The GMM entropy lacks a closed-form so NPV applies Jensen's lower bound as an approximation, $\hat{H}_q^J(q(x))$. The cross entropy also lacks a closed-form, so NPV approximates this term using the analogous Huber et al. Taylor approximation. Specifically, NPV expands the log density around the means of each GMM component as,

$$H_q(p(x)) \approx - \sum_{i=1}^{M} w_i \sum_{n=0}^{N} \frac{\nabla^2 \log(p(\mu_i))}{n!} \mathbb{E}_{q_i} \left[ (x - \mu_i)^n \right] = \hat{H}_{N,q}^H(p(x)) \tag{20}$$

However, Eqn. (20) is subject to the divergence criterion of Theorem 3.1 if $2p(\mu_i) \leq \max(p(x))$. By replacing the entropy terms with our convergent series approximations we observe significant improvements in accuracy.

**In our approach**, we will highlight and address two problems with the NPV algorithm; the potential divergence of $\hat{H}_{N,q}^H(p(x))$ and the poor estimation of the GMM entropy via $\hat{H}_q^J(q(x))$. To address the potential divergence of $\hat{H}_{N,q}^H(p(x))$, we will take motivation from the results found in [23] and use a 2 point Gauss-Hermite quadrature method to approximate $H_q(p(x))$. This method will be a limiting factor in scaling the NPV algorithm in dimension, however it guarantees that the cross-entropy approximation will not diverge. This alteration leads to a solution for the inconsistency of the ELBO approximations. Then, Jensen's inequality is a very poor approximation for entropy in general, instead we will use the three methods we have introduced, our Taylor, Taylor limit, and our Legendre, as the GMM entropy approximations for higher accuracy. Fig. 4 shows an approximation of a two dimensional, three component mixture Student T distribution using a five component GMM in the traditional NPV, our modified NPV algorithm with our Taylor and Legendre approximation.

**The results**, as seen in Fig. 5, highlight the accuracy of each method versus the number of components, the order of our polynomial approximation, and the dimension of the GMM. In each experiment, we are approximating a multivariate mixture T distribution, $p(x)$. We randomize the parameters of $p(x)$ and the initialization parameters of the variational GMM, $q(x)$, for optimization. The KL is approximated using a 100000 Monte Carlo approximation after convergence of each algorithm. We see that in all cases of components, order, and dimension, our method achieves significant accuracy improvements. We see that we can use low order approximations to receive substantial approximation improvement (Fig. 5 (middle)). We see all methods gain accuracy as number of components increase (Fig. 5 (left)) however our methods see most of the accuracy improvements with only a few components, whereas NPV has substantially worse approximations with low components. Finally, we see we maintain a lower variance and KL than NPV with all our methods as the dimension grows (Fig. 5 (right)). For further discussion of the experiment, see A.6.

## 7   Limitations

Our Taylor and Legendre methods ensure convergence and deliver comparable accuracy to that of Huber et al. However, the computational complexity of these methods grows with the number of

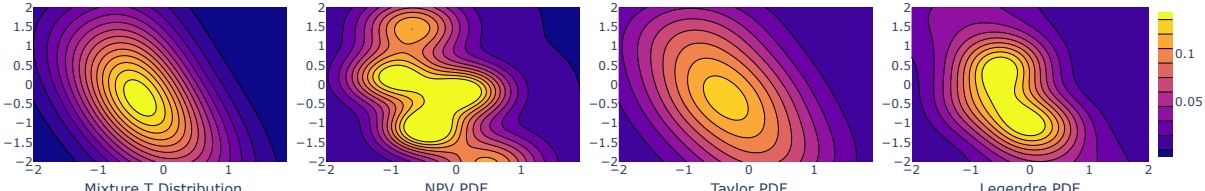

Figure 4: A three component mixture Student T distribution PDF (far-left) is approximated by a five component GMM using traditional NPV (left), our algorithm using a $6^{th}$ order Taylor polynomial (right), and Legendre polynomial (far-right). We see that NPV both has issues with finding correct placement of means and sets the variances of the GMM components to be too narrow. Our methods do a better job of assigning means and the Legendre method seems to set the variances slightly better than our Taylor.

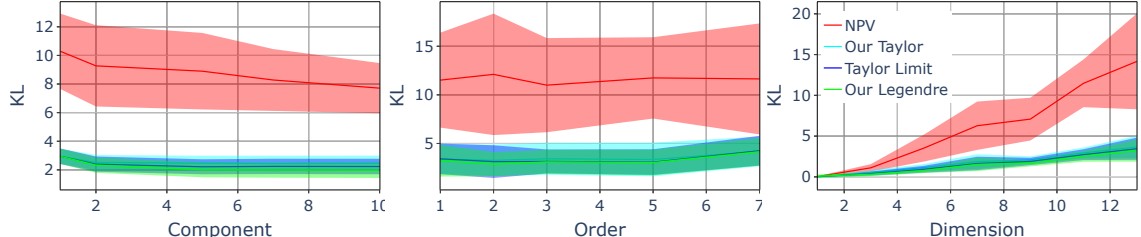

Figure 5: The above figures show the accuracy of each method across varying components, orders, and dimensions in approximating a multivariate mixture T distribution with a GMM. Our method consistently improves accuracy significantly. Low order of the convergent estimators provide substantial approximation improvement (middle). Most accuracy improvements are achieved with a small number of components, unlike NPV (left) which continues to need higher number of components to see good accuracy return. The guaranteed convergence of the approximation in higher dimensions seems to have a drastic improvement on accuracy ().

components, $M$, in the GMM, following an $\mathcal{O}((n + M - 1)!)$ time complexity, where $n$ represents the order of the approximating polynomial. This summation comprises only scalar terms and is amenable to parallelization but may become prohibitively expensive for large $M$ and $n$ values. Furthermore, our polynomial methods are specifically tailored to the entropy, $H_p(p(x))$, and cross-entropy, $H_p(q(x))$, where both $p(x)$ and $q(x)$ are GMMs. In contrast, Huber et al.'s approximation can be more readily extended to cross-entropy scenarios where $q(x)$ pertains to any distribution amenable to the construction of a second-order Taylor polynomial. This limitation hinders the broader applicability of our approximation.

# 8 Discussion

We have provided novel theoretical analysis of the convergence for the widely used Huber et al. Taylor approximation of GMM entropy and established that the series diverges under conditions on the component means. We address this divergence by introducing multiple novel methods which provably converge. We wish to emphasize that the Huber et al. approximation tends to yield accurate results when it is convergent and the intention of this work is not to dissuade the use of this approximator. Quite the contrary, this work encourages the use of either Huber et al. or our own estimator by providing a solid theoretical foundation for both methods. We acknowledge that there are contexts in which one method may be preferred over the other, for example when bounds are preferred, or when convergence criteria are provably satisfied.

There are several areas that require further investigation. For example, one limitation of both methods is that they scale poorly with polynomial order and number of components. In fact, Huber et al. cannot easily be calculated for fourth order and above, due to tensor arithmetic. Our approximation works well in practice, but is limited solely to GMM densities. Further work is necessary to efficiently apply our convergent series to situations of cross-entropy's that contain non-GMM distributions.

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

# A   Appendix

## A.1   Section 3 Proofs

**Theorem 3.1** (Divergence Criterion for Huber et al.). *Let $p(x) = \sum_{i=1}^{M} w_i \mathcal{N}(x \mid \mu_i, \Sigma_i)$ be a GMM and consider the Taylor series presented by Huber et al. in Eqn. (5). If any mean component, $\mu_i$, satisfies the condition $p(\mu_i) < \frac{1}{2} max(p(x))$, then Huber et al.'s approximation diverges, otherwise it converges.*

*Proof.* Let $f(y) = \log(y)$, $g(x) = p(x)$, and $h(x) = f(g(x)) = \log(p(x))$. Huber et al. creates the Taylor series in Eqn. (5) with $N^{th}$ order approximation

$$\sum_{i=1}^{M} w_i T_{h,N,\mu_i}(x) = \sum_{i=1}^{M} w_i \sum_{n=0}^{N} \frac{h^{(n)}(\mu_i)}{n!}(x - \mu_i)^n, \tag{21}$$

Let us consider just a single one of the Taylor series in Eqn. (21)

$$T_{h,N,\mu_i}(x) = \sum_{n=0}^{N} \frac{h^{(n)}(\mu_i)}{n!}(x - \mu_i)^n, \tag{22}$$

By Theorem 3.4 in Lang[1], the Taylor series of a composition of function is equivalent to the composition of each components Taylor series, i.e., $T_{h,N,\mu_i}(x) = T_{f,N,g(\mu_i)} \circ T_{g,N,\mu_i}(x)$ where $\circ$ is the composition operation. Since $p(x)$ is a GMM, is the sum of entire functions, and thus itself is entire, meaning it's Taylor series, $T_{g,N,\mu_i}(x)$, converges everywhere. We turn our attention to the Taylor approximation of log, $T_{f,N,g(\mu_i)}$,

$$T_{f,N,p(\mu_i)}(x) = \log(p(\mu_i)) + \sum_{n=1}^{N} \frac{(-1)^{n-1}}{n p(\mu_i)^n}(y - p(\mu_i))^n, \tag{23}$$

We can look at the $n^{th}$ term in the series, $b_n = \frac{(-1)^{n-1}}{n p(\mu_i)^n}(y - p(\mu_i))^n$, and use the ratio test to define convergence.

$$\begin{aligned}
\lim_{n \to \infty} \left| \frac{b_{n+1}}{b_n} \right| &= \lim_{n \to \infty} \left| \frac{(-1)^n}{(n+1)p(\mu_i)^{n+1}}(y - p(\mu_i))^{n+1} \frac{n p(\mu_i)^n}{(-1)^{n-1}}(y - p(\mu_i))^{-n} \right| \\
&= \lim_{n \to \infty} \left| \frac{(-1)^n}{(-1)^{n-1}} \frac{n p(\mu_i)^n}{(n+1)p(\mu_i)^{n+1}} \frac{(y - p(\mu_i))^{n+1}}{(y - p(\mu_i))^n} \right| \\
&= \lim_{n \to \infty} \left| \frac{n}{(n+1)p(\mu_i)}(y - p(\mu_i)) \right| \\
&= \lim_{n \to \infty} \frac{n}{(n+1)} \left| \frac{y - p(\mu_i)}{p(\mu_i)} \right| = \left| \frac{y - p(\mu_i)}{p(\mu_i)} \right| = L
\end{aligned}$$

The ratio test states that the series converges if the limit, $L$, is strictly less than 1. However, setting $L = 1$ and some simple manipulation, we find

$$\left| \frac{y - p(\mu_i)}{p(\mu_i)} \right| < 1 \quad \Rightarrow \quad |y - p(\mu_i)| < p(\mu_i) \quad \Rightarrow \quad y < 2p(\mu_i) \tag{24}$$

This only converges if all $y < 2(p(\mu_i))$. Consider the maximum, $max(p(x))$, if it satisfies this condition, so will every other point, if it doesn't satisfy this point, then the series is divergent by the ratio test. So we have the convergent criterion $max(p(x)) < 2p(\mu_i)$, or written in terms of divergence criterion, we diverge if $p(\mu_i) < \frac{1}{2}max(p(x))$. $\qquad \square$

**Lemma 3.2** (Convergent Taylor Series of Log). *If $a > \frac{1}{2}max(p(x))$, then for all $x$*

$$\log(p(x)) = \log(a) + \sum_{n=1}^{\infty} \frac{(-1)^{n-1}}{n a^n} \sum_{k=0}^{n} \binom{n}{k}(-a)^{n-k} p(x)^k \tag{25}$$

---

[1]Lang, Serge. Complex Analysis. 4th ed. Springer, 2013. ISBN 978-1-4757-3083-8.

*Proof.* Consider the $n^{th}$ term in the sum

$$b_n = \frac{(-1)^{n-1}}{na^n}(p(x) - a)^n$$

The ratio test says if the limit of the absolute value of successive terms converges to a value strictly less than one, then the series converges

$$\lim_{n\to\infty}\left|\frac{b_{n+1}}{b_n}\right| = \lim_{n\to\infty}\left|\frac{(-1)^n}{(n+1)a^{n+1}}(p(x) - a)^{n+1}\frac{na^n}{(-1)^{n-1}}(p(x) - a)^{-n}\right|$$

$$= \lim_{n\to\infty}\left|\frac{(-1)^n}{(-1)^{n-1}}\frac{na^n}{(n+1)a^{n+1}}\frac{(p(x) - a)^{n+1}}{(p(x) - a)^n}\right|$$

$$= \lim_{n\to\infty}\left|\frac{n}{(n+1)a}(p(x) - a)\right|$$

$$= \lim_{n\to\infty}\frac{n}{(n+1)}\left|\frac{p(x) - a}{a}\right| = \left|\frac{p(x) - a}{a}\right| = L$$

We see that $L < 1 \ \forall x$ iff $a > \frac{1}{2}\max(p(x))$ in which case the series converges everywhere. $\qquad\square$

**Lemma 3.3** (Convergent Legendre Series of Log). *If $a > max(p(x))$, and consider the $n^{th}$ shifted Legendre polynomial on the interval $[0, a]$ in Eqn. (8). Then for $x$ a.e.*

$$\log(p(x)) = \sum_{n=0}^{\infty}(2n + 1)\sum_{j=0}^{n}\frac{(-1)^{n+j}(n + j)!((j + 1)\log(a) - 1)}{(n - j)!((j + 1)!)^2}L_{[0,a],n}(p(x)) \qquad (26)$$

*Proof.* Orthogonal polynomials can approximate any function a.e. that is continuous and square-integrable (see Trefethen and Bau[2] and Gustafson[3]). In out case, $L_{[0,a],n}(y)$ live on $L_2([0, a])$ (referring to the second Lebesgue space on the interval $[0, a]$). This means all we have to show is that $\log(y)$ lives in this domain which means $\|\log(y)\|_2^2 < \infty$

$$\|\log(y)\|_2^2 = \int_0^a \log(y)^2 dy = a((\log(a) - 2)\log(a) + 2) < \infty \qquad (27)$$

So we see that $\log(y) \in L_2([0, a])$ and therefore it's Legendre series is convergent.

For completeness, we now derive the Legendre series for $\log(y)$. We will appeal to Eqn. (8), $L_{[0,a],n} = \sum_{k=0}^{n}\frac{(-1)^{n+k}(n + k)!}{(n - k)!(k!)^2a^k}y^k$, and $\langle L_{[0,a],n}(y), L_{[0,a],n}(y)\rangle = \frac{a}{2n+1}$ as found in [4]

$$\log(p(x)) = \sum_{n=0}^{\infty}\frac{\langle\log(y), L_{[0,a],n}(y)\rangle}{\langle L_{[0,a],n}(y), L_{[0,a],n}(y)\rangle}L_{[0,a],n}(p(x))$$

$$= \sum_{n=0}^{\infty}\frac{2n + 1}{a}\int_0^a \log(y)\sum_{k=0}^{n}\frac{(-1)^{n+k}(n + k)!}{(n - k)!(k!)^2a^k}y^k dy L_{[0,a],n}(p(x))$$

$$= \sum_{n=0}^{\infty}\frac{2n + 1}{a}\sum_{k=0}^{n}\frac{(-1)^{n+k}(n + k)!}{(n - k)!(k!)^2a^k}\int_0^a \log(y)y^k dy L_{[0,a],n}(p(x))$$

$$= \sum_{n=0}^{\infty}\frac{2n + 1}{a}\sum_{k=0}^{n}\frac{(-1)^{n+k}(n + k)!}{(n - k)!(k!)^2a^k}\frac{a^{k+1}((k + 1)\log(a) - 1)}{(k + 1)^2}L_{[0,a],n}(p(x))$$

$$= \sum_{n=0}^{\infty}2n + 1\sum_{k=0}^{n}\frac{(-1)^{n+k}(n + k)!((k + 1)\log(a) - 1)}{(n - k)!((k + 1)!)^2}L_{[0,a],n}(p(x))$$

which we know is convergent from the above discussion $\qquad\square$

---

[2]Trefethen, Lloyd N., and David Bau III. Numerical Linear Algebra. SIAM, 1997.

[3]Grant B. Gustafson Differential Equations and Linear Algebra. 1999-2022.

## A.2 Section 4 Proofs

**Lemma 4.1** (Closed form expectation of powers of GMMs). *Let $p(x) = \sum_{i=1}^{M} w_i \mathcal{N}(x|\mu_i, \Sigma_i)$ be a GMM and $k$ be a non-negative integer. Then*

$$\mathbb{E}_p[p(x)^k] = \sum_{j_1 + \cdots + j_M = k} \binom{k}{j_1, \ldots, j_M} \sum_{i=1}^{M} w_i \left( \frac{\mathcal{N}(0|\mu_i, \Sigma_i)}{\mathcal{N}(0|\mu, \Sigma)} \prod_{t=1}^{M} (w_t \mathcal{N}(0|\mu_t, \Sigma_t)^{j_t}) \right) \quad (28)$$

*where $\Sigma = (\Sigma_i^{-1} + \sum_{t=1}^{M} j_t \Sigma_t^{-1})^{-1}$ and $\mu = \Sigma(\Sigma_i^{-1}\mu_i + \sum_{t=1}^{M} j_t \Sigma_t^{-1}\mu_t)$.*

*Proof.* We will prove this statement by directly expanding out each term

$$\mathbb{E}_p[p(x)^k] = \mathbb{E}_p \left[ \left( \sum_{t=1}^{m} w_t \mathcal{N}(x|\mu_t, \Sigma_t) \right)^k \right]$$

$$= \mathbb{E}_p \left[ \sum_{j_1 + \cdots + j_m = k} \binom{k}{j_1, \ldots, j_m} \prod_{t=1}^{m} (w_t \mathcal{N}(x|\mu_t, \Sigma_t))^{j_t} \right]$$

$$= \sum_{j_1 + \cdots + j_m = k} \binom{k}{j_1, \ldots, j_m} \prod_{t'=1}^{m} (w_{t'})^{j_{t'}} \sum_{i=1}^{M} w_i \int \mathcal{N}(x|\mu_i, \Sigma_i) \prod_{t=1}^{m} (\mathcal{N}(x|\mu_t, \Sigma_t))^{j_t} dx$$

To combine the Gaussians under the integral, we appeal to the power of Gaussians (Lemma A.2.3) and product of Gaussians (Lemma A.2.4)

$$= \sum_{j_1 + \cdots + j_m = k} \binom{k}{j_1, \ldots, j_m} \prod_{t'=1}^{m} (w_{t'})^{j_{t'}} \sum_{i=1}^{M} w_i \int \mathcal{N}(x|\mu_i, \Sigma_i) \prod_{t=1}^{m} \frac{\mathcal{N}(x|\mu_t, \frac{1}{j_t}\Sigma_t)}{|j_t (2\pi\Sigma_t)^{j_t - 1}|^{1/2}} dx$$

$$= \sum_{j_1 + \cdots + j_m = k} \binom{k}{j_1, \ldots, j_m} \prod_{t'=1}^{m} \frac{(w_{t'})^{j_{t'}}}{|j_t (2\pi\Sigma_t)^{j_t - 1}|^{1/2}} \sum_{i=1}^{M} w_i \int \frac{\mathcal{N}(0|\mu_i, \Sigma_i) \prod_{t=1}^{m} \mathcal{N}(0|\mu_t, \frac{1}{j_t}\Sigma_t)}{\mathcal{N}(0|\mu, \Sigma)} N(x|\mu, \Sigma) dx$$

$$= \sum_{j_1 + \cdots + j_m = k} \binom{k}{j_1, \ldots, j_m} \sum_{i=1}^{M} w_i \left( \frac{\mathcal{N}(0|\mu_i, \Sigma_i)}{\mathcal{N}(0|\mu, \Sigma)} \prod_{t=1}^{m} (w_t \mathcal{N}(0|\mu_t, \Sigma_t))^{j_t} \right)$$

where $\mu = \Sigma(\Sigma_i^{-1}\mu_i + \sum_{t=1}^{M} j_t \Sigma_t^{-1}\mu_t)$ as defined from Lemma A.2.4. We see that we are left with no integral and a closed form of the expectation of the powers of the GMM. $\square$

**Theorem 4.2** (Convergence of $\hat{H}_{N,a}^T(p(x))$). *Let $p(x) = \sum_{i=1}^{M} w_i \mathcal{N}(x|\mu_i, \Sigma_i)$ be a GMM and choose a Taylor center such that $a > \frac{1}{2}\max(p(x))$. Then, for $\hat{H}_{N,a}^T(p(x))$ defined in Eqn. (12)*

$$\lim_{N \to \infty} \hat{H}_{N,a}^T(p(x)) = H(p(x)) \quad (29)$$

*Proof.* We start out with the definition of entropy and introduce in the approximation discussed in Lemma 3.2

$$H(p(x)) = - \int \sum_{i=1}^{M} w_i q_i(x) \log(p(x)) dx = - \sum_{i=1}^{M} w_i \int q_i(x) \log(p(x)) dx$$

$$= - \sum_{i=1}^{M} w_i \int q_i(x) \left( \log(a) + \sum_{i=1}^{\infty} \frac{(-1)^{n-1}}{na^n} (p(x) - a)^n \right) dx$$

$$= - \sum_{i=1}^{M} w_i \left( \log(a) + \int q_i(x) \sum_{i=1}^{\infty} \frac{(-1)^{n-1}}{na^n} (p(x) - a)^n dx \right)$$

$$= - \sum_{i=1}^{M} w_i \left( \log(a) + \int \sum_{i=1}^{\infty} \frac{(-1)^{n-1}}{na^n} q_i(x)(p(x) - a)^n dx \right)$$

We now wish to swap the order of integration and of the infinite summation as shown in Lemma A.2.1

$$= -\sum_{i=1}^{M} w_i \left( log(a) + \sum_{i=1}^{\infty} \int \frac{(-1)^{n-1}}{na^n} q_i(x)(p(x)-a)^n dx \right)$$

$$= -\sum_{i=1}^{M} w_i \left( log(a) + \sum_{i=1}^{\infty} \frac{(-1)^{n-1}}{na^n} \int q_i(x)(p(x)-a)^n dx \right)$$

$$= -\sum_{i=1}^{M} w_i \left( log(a) + \sum_{i=1}^{\infty} \frac{(-1)^{n-1}}{na^n} \int q_i(x) \sum_{k=0}^{n} \binom{n}{k}(-a)^{n-k}(p(x))^k dx \right)$$

$$= -\sum_{i=1}^{M} w_i \left( log(a) + \sum_{i=1}^{\infty} \frac{(-1)^{n-1}}{na^n} \sum_{k=0}^{n} \binom{n}{k}(-a)^{n-k} \mathbb{E}_{q_i(x)}\left[(p(x))^k\right] \right)$$

We can compute $\mathbb{E}_{q_i(x)}\left[(p(x))^k\right]$ using Lemma 4.1. The above term is equality for the Entropy, simply truncating the infinite summation gives a convergent approximation.

$$\hat{H}_{N,a}^{T}(p(x)) = -\sum_{i=1}^{M} w_i \left( log(a) + \sum_{i=1}^{N} \frac{(-1)^{n-1}}{na^n} \sum_{k=0}^{n} \binom{n}{k}(-a)^{n-k} \mathbb{E}_{q_i(x)}\left[(p(x))^k\right] \right) \quad (30)$$

$\square$

**Theorem 4.3** (Taylor Series is Lower Bound of Entropy). *Let $p(x) = \sum_{i=1}^{M} w_i \mathcal{N}(x|\mu_i, \Sigma_i)$ and $a > max(p(x))$. Then, for all finite $N$,*

$$\hat{H}_{N,a}^{T}(p(x)) \leq H(p(x)) \quad (31)$$

*Proof.* If $a > max(p(x))$, then we have the following lower bound

$$H(p(x)) = -\int p(x) \log p(x) dx = -\int p(x) \left( \log(a) + \sum_{n=1}^{\infty} \frac{(-1)^{n-1}}{na^n}(p(x)-a)^n \right) dx$$

$$= -\int p(x) \left( \log(a) + \sum_{n=1}^{\infty} \frac{(-1)^{n-1}(-1)^n}{na^n}(a-p(x))^n \right) dx$$

$$= -\log(a) - \int p(x) \left( \sum_{n=1}^{\infty} \frac{-1}{na^n}(a-p(x))^n \right) dx$$

$$= -\log(a) + \int p(x) \left( \sum_{n=1}^{\infty} \frac{1}{na^n}(a-p(x))^n \right) dx$$

$$\geq -\log(a) + \int p(x) \left( \sum_{n=1}^{N} \frac{1}{na^n}(a-p(x))^n \right) dx = \hat{H}_N(p(x))$$

since every term in the summation is positive due to $a > p(x) \; \forall x$, then truncating the series only removes positive terms, leaving us with a lower bound. $\square$

**Theorem 4.4** (Upper bound on maximum of a GMM). *Let $p(x) = \sum_{i=1}^{M} w_i \mathcal{N}(x|\mu_i, \Sigma_i)$, then*

$$max(p(x)) \leq a = \sum_{i}^{M} w_i \left|2\pi\Sigma_i\right|^{-\frac{1}{2}} \quad (32)$$

*Proof.* We need to find an upper bound on $max(p(x))$

$$max(p(x)) = max\left( \sum_{i=1}^{M} w_i \mathcal{N}(x|\mu_i, \Sigma_i) \right)$$

$$\leq \sum_{i=1}^{M} w_i max\left( \mathcal{N}(x|\mu_i, \Sigma_i) \right) = \sum_{i=1}^{M} w_i \left|2\pi\Sigma_i\right|^{-\frac{1}{2}}$$

We simply have bound the maximum of the combination by combining the maximum of every component in the GMM. □

**Theorem 4.5** (Convergence of $\hat{H}^L_{N,a}(p(x))$). *Let $p(x) = \sum_{i=1}^M w_i \mathcal{N}(x|\mu_i, \Sigma_i)$ be a GMM and choose an interval such that $a > max(p(x))$. Then for $\hat{H}^L_{N,a}(p(x))$ defined in Eqn.* (16)

$$\lim_{N \to \infty} \hat{H}^L_{N,a}(p(x)) = H(p(x)) \tag{33}$$

*Proof.* We start out with the definition of entropy and introduce in the approximation discussed in Lemma 3.3

$$H(p(x)) = -\int \sum_{i=1}^M w_i q_i(x) \log(p(x)) dx = -\sum_{i=1}^M w_i \int q_i(x) \log(p(x)) dx$$

$$= -\sum_{i=1}^M w_i \int q_i(x) \sum_{i=0}^{\infty} \frac{\langle \log(y), L_{[0,a],n}(y) \rangle}{\langle L_{[0,a],n}(y), L_{[0,a],n}(y) \rangle} L_{[0,a],n}(p(x)) dx$$

$$= -\sum_{i=1}^M w_i \sum_{i=0}^{\infty} \int q_i(x) \frac{\langle \log(y), L_{[0,a],n}(y) \rangle}{\langle L_{[0,a],n}(y), L_{[0,a],n}(y) \rangle} L_{[0,a],n}(p(x)) dx$$

We swapped the order of integration and of the infinite summation as shown in Lemma A.2.2. We simplify computation that are recreated in Theorem 3.3

$$= -\sum_{i=0}^{\infty} \frac{\langle \log(y), L_{[0,a],n}(y) \rangle}{\langle L_{[0,a],n}(y), L_{[0,a],n}(y) \rangle} \sum_{i=1}^M w_i \int q_i(x) L_{[0,a],n}(p(x)) dx$$

$$= -\sum_{i=0}^{\infty} (2n+1) \sum_{j=0}^n \frac{(-1)^{n+j}(n+j)!((j+1)\log(a)-1)}{(n-j)!((j+1)!)^2} \sum_{i=1}^M w_i \mathbb{E}_{q_i(x)} \left[ \sum_{k=0}^n \frac{(-1)^{n+k}(n+k)!}{(n-k)!(k!)^2 a^k} p(x)^k \right]$$

$$= -\sum_{i=0}^{\infty} (2n+1) \sum_{j=0}^n \frac{(-1)^{n+j}(n+j)!((j+1)\log(a)-1)}{(n-j)!((j+1)!)^2} \sum_{k=0}^n \frac{(-1)^{n+k}(n+k)!}{(n-k)!(k!)^2 a^k} \sum_{i=1}^M w_i \mathbb{E}_{q_i(x)} \left[ p(x)^k \right]$$

We can compute $\mathbb{E}_{q_i(x)} \left[ (p(x))^k \right]$ using Lemma 4.1. Simply truncating the infinite summation gives the approximation.

$$\hat{H}^L_{N,a}(p(x)) = -\sum_{i=0}^N (2n+1) \sum_{j=0}^n \frac{(-1)^{n+j}(n+j)!((j+1)\log(a)-1)}{(n-j)!((j+1)!)^2} \sum_{k=0}^n \frac{(-1)^{n+k}(n+k)!}{(n-k)!(k!)^2 a^k} \sum_{i=1}^M w_i \mathbb{E}_{q_i(x)} \left[ p(x)^k \right]$$

which $\lim_{N \to \infty} \hat{H}^L_{N,a}(p(x)) = H(p(x))$ as the above series is exactly equal to the entropy. □

Here, we address a few of the assumptions we made in the above derivations. We start with the ability to swap the order of the integral and infinite sum for the Taylor series.

**Lemma A.2.1** (Swapping Integral and Infinite Sum (Taylor)). *Let $a > \frac{1}{2}max(p(x))$, then*

$$\int \sum_{i=1}^{\infty} \frac{(-1)^{n-1}}{na^n} q_i(x)(p(x)-a)^n dx = \sum_{i=1}^{\infty} \int \frac{(-1)^{n-1}}{na^n} q_i(x)(p(x)-a)^n dx$$

*Proof.* For simplicity of notation, let $c = \sup \left| \frac{p(x)-a}{a} \right| < 1$ since $a > \frac{1}{2}\max(p(x))$. We then appeal to Fubini-Tonelli theorem which states that if $\int \sum |f_n(x)| dx < \infty$ or if $\sum \int |f_n(x)| dx < \infty$, then $\int \sum f_n(x) dx = \sum \int f_n(x) dx$.

$$\sum_{i=1}^{\infty} \int \left| \frac{(-1)^{n-1}}{na^n} q_i(x)(p(x)-a)^n \right| dx = \sum_{i=1}^{\infty} \frac{1}{n} \int q_i(x) \left( \frac{|p(x)-a|}{a} \right)^n dx$$

$$\leq \sum_{i=1}^{\infty} \frac{1}{n} \int q_i(x) (c)^n dx$$

$$= \sum_{i=1}^{\infty} \frac{c^n}{n} \int q_i(x) dx = \sum_{i=1}^{\infty} \frac{c^n}{n} < \infty$$

We know that $\sum_{i=1}^{\infty} \frac{c^n}{n} < \infty$ because of the ratio test again

$$\lim_{n\to\infty}\left|\frac{c^{n+1}}{n+1}\frac{n}{c^n}\right| = \lim_{n\to\infty}\frac{n}{n+1}c = c < 1$$

So we see that Fubini-Tonelli holds so

$$\int \sum_{i=1}^{\infty} \frac{(-1)^{n-1}}{na^n} q_i(x)(p(x)-a)^n dx = \sum_{i=1}^{\infty} \int \frac{(-1)^{n-1}}{na^n} q_i(x)(p(x)-a)^n dx$$

$\square$

We now consider the case for the Legendre series

**Lemma A.2.2** (Swapping Integral and Infinite Sum (Legendre)). *Let $a > max(p(x))$, then*

$$\int q_i(x) \sum_{i=1}^{\infty} \frac{\langle \log(y), L_{[0,a],n}(y)\rangle}{\langle L_{[0,a],n}(y), L_{[0,a],n}(y)\rangle} L_{[0,a],n}(p(x))dx = \sum_{i=1}^{\infty} \int q_i(x) \frac{\langle \log(y), L_{[0,a],n}(y)\rangle}{\langle L_{[0,a],n}(y), L_{[0,a],n}(y)\rangle} L_{[0,a],n}(p(x))dx$$

*Proof.* We again appeal to Fubini-Tonelli. We will use that $\left|\frac{L_{[0,a],n}(p(x))}{\langle L_{[0,a],n}(y), L_{[0,a],n}(y)\rangle}\right| \leq 1$ as it is the orthonormal polynomials and then use Cauchy-Schwartz on the remaining term

$$\sum_{i=1}^{\infty} \int \left| q_i(x) \frac{\langle \log(y), L_{[0,a],n}(y)\rangle}{\langle L_{[0,a],n}(y), L_{[0,a],n}(y)\rangle} L_{[0,a],n}(p(x)) \right| dx$$

$$= \sum_{i=1}^{\infty} \langle \log(y), L_{[0,a],n}(y)\rangle \int \left| \frac{q_i(x)L_{[0,a],n}(p(x))}{\langle L_{[0,a],n}(y), L_{[0,a],n}(y)\rangle} \right| dx$$

$$\leq \sum_{i=1}^{\infty} \|\log(y)\|^2 \|L_{[0,a],n}(y)\|^2 \int q_i(x)dx$$

$$= \sum_{i=1}^{\infty} a((\log(a)-2)\log(a)+2)\left(\frac{a}{2n+1}\right)^2 \int N(x|\mu_i, \Sigma_i)dx$$

$$= a^3((\log(a)-2)\log(a)+2)\sum_{i=1}^{\infty} \frac{1}{(2n+1)^2} < \infty$$

So we see that since the absolute value is finite, then Fubini-Tonelli applies and we can swap the order of the integral and infinite summation. $\square$

The next thing we show is the relations we used for powers of Gaussians.

**Lemma A.2.3.** *Powers of a Gaussian*

$$\mathcal{N}(x|\mu, \Sigma)^n = \left|n(2\pi\Sigma)^{n-1}\right|^{-\frac{1}{2}} \mathcal{N}\left(x|\mu, \frac{1}{n}\Sigma\right)$$

*Proof.* We are going to do an inductive proof and rely on the well known relation of products of Gaussians

$$\mathcal{N}(x|a, A)\mathcal{N}(x|b, B) = \mathcal{N}(x|d, D)\mathcal{N}(a|b, A+B)$$

Where $D = (A^{-1}+B-1)^{-1}$ and $d = D(A^{-1}a + B^{-1}b)$.

**Base Case:** $n = 2$

$$\mathcal{N}(x|\mu, \Sigma)^2 = \mathcal{N}(x|\mu, \Sigma)\mathcal{N}(x|\mu, \Sigma) \tag{34}$$

$$= N(x|\mu, \frac{1}{2}\Sigma)N(\mu|\mu, 2\Sigma) = |2\pi\Sigma|^{-\frac{1}{2}} N(x|\mu, \frac{1}{2}\Sigma) \tag{35}$$

In this case, we get that $D = (\Sigma^{-1} + \Sigma-1)^{-1} = \frac{1}{2}\Sigma$ and $d = \frac{1}{2}\Sigma(\Sigma^{-1}\mu + \Sigma^{-1}\mu) = \mu$. We also see that $N(\mu|\mu, 2\Sigma)$ is being evaluated at its maximum, which just leaves the scaling term out front

of the exponential in the Gaussian, $|2\pi\Sigma|^{-\frac{1}{2}}$.

**Inductive step:** Assume that $N(x|\mu,\Sigma)^n = \left|n(2\pi\Sigma)^{n-1}\right|^{-\frac{1}{2}} \mathcal{N}\left(x|\mu, \frac{1}{n}\Sigma\right)$, then we wish to show that $N(x|\mu,\Sigma)^{n+1} = \left|(n+1)(2\pi\Sigma)^n\right|^{-\frac{1}{2}} \mathcal{N}\left(x|\mu, \frac{1}{n+1}\Sigma\right)$

$$
\begin{aligned}
N(x|\mu,\Sigma)^{n+1} &= N(x|\mu,\Sigma)^n N(x|\mu,\Sigma) \\
&= \left|n(2\pi\Sigma)^{n-1}\right|^{-\frac{1}{2}} \mathcal{N}\left(x|\mu, \frac{1}{n}\Sigma\right) N(x|\mu,\Sigma) \\
&= \left|n(2\pi\Sigma)^{n-1}\right|^{-\frac{1}{2}} \mathcal{N}\left(x|\mu, \frac{1}{n+1}\Sigma\right) N(\mu|\mu, \frac{n+1}{n}\Sigma) \\
&= \left|n(2\pi\Sigma)^{n-1}\right|^{-\frac{1}{2}} \left|2\pi\frac{n+1}{n}\Sigma\right|^{-\frac{1}{2}} \mathcal{N}\left(x|\mu, \frac{1}{n+1}\Sigma\right) \\
&= \left|(n+1)(2\pi\Sigma)^n\right|^{-\frac{1}{2}} \mathcal{N}\left(x|\mu, \frac{1}{n+1}\Sigma\right)
\end{aligned}
$$

Here, $D = \left(\left(\frac{1}{n}\Sigma\right)^{-1} + \Sigma^{-1}\right)^{-1} = \frac{1}{n+1}\Sigma$ and $d = \frac{1}{n+1}\Sigma\left(\left(\frac{1}{n}\Sigma\right)^{-1}\mu + \Sigma^{-1}\mu\right) = \mu$. $\qquad\square$

We finally show the product of Gaussians that we used. We keep the exact same notation used in the derivation of the entropy Taylor series so the terms may be more easily identified.

**Lemma A.2.4.** *Product of a Gaussians*

$$
\mathcal{N}(x|\mu_i,\Sigma_i)\prod_{t=1}^{m}\mathcal{N}\left(x|\mu_t, \frac{1}{j_t}\Sigma_t\right) = \mathcal{N}(0|\mu_i,\Sigma_i)\prod_{t=1}^{m}\mathcal{N}\left(0+\mu_t, \frac{1}{j_t}\Sigma_t\right)\frac{N(x|\mu,\Sigma)}{N(0|\mu,\Sigma)}
$$

*where* $\Sigma = (\Sigma_i^{-1} + \sum_{t=1}^{m}j_t\Sigma_t^{-1})^{-1}$ *and* $\mu = \Sigma\left(\Sigma_i^{-1}\mu_i + \sum_{t=1}^{m}j_t\Sigma_t^{-1}\mu_t\right)$

*Proof.* We will simply expand out the product of Gaussians, collect like terms, complete the square, and then recollect exponentials into Gaussians evaluated at 0.

$$
\mathcal{N}(x|\mu_i,\Sigma_i)\prod_{t=1}^{m}\mathcal{N}\left(x|\mu_t, \frac{1}{j_t}\Sigma_t\right) = |2\pi\Sigma_i|^{-\frac{1}{2}}\exp\left\{-\frac{1}{2}(x-\mu_i)^T\Sigma_i^{-1}(x-\mu_i)\right\}\prod_{t=1}^{m}\left|\frac{2\pi}{j_t}\Sigma_t\right|^{-\frac{1}{2}}\exp\left\{-\frac{1}{2}(x-\mu_t)^T j_t\Sigma_t^{-1}(x-\mu_t)\right\}
$$

$$
= |2\pi\Sigma_i|^{-\frac{1}{2}}\prod_{t=1}^{m}\left|\frac{2\pi}{j_t}\Sigma_t\right|^{-\frac{1}{2}}\exp\left\{-\frac{1}{2}\left(x^T\Sigma_i^{-1}x - 2x^T\Sigma_i^{-1}\mu_i + \mu_i^T\Sigma_i^{-1}\mu_i + \sum_{t=1}^{m}x^T j_t\Sigma_t^{-1}x - 2x^T j_t\Sigma_t^{-1}\mu_t + \mu_t^T j_t\Sigma_t^{-1}\mu_t\right)\right\}
$$

$$
= |2\pi\Sigma_i|^{-\frac{1}{2}}\prod_{t=1}^{m}\left|\frac{2\pi}{j_t}\Sigma_t\right|^{-\frac{1}{2}}\exp\left\{-\frac{1}{2}\left(x^T\left(\Sigma_i^{-1} + \sum_{t=1}^{m}j_t\Sigma_t\right)x - 2x^T\left(\Sigma_i^{-1}\mu_i + \sum_{t=1}^{m}j_t\Sigma_t^{-1}\right) + \mu_i^T\Sigma_i^{-1}\mu_i + \sum_{t=1}^{m}\mu_t^T j_t\Sigma_t^{-1}\mu_t\right)\right\}
$$

Now we let $\Sigma = (\Sigma^{-1} + \sum_{t=1}^{m}j_t\Sigma_t^{-1})^{-1}$ and $\mu = \Sigma\left(\Sigma_i^{-1}\mu_i + \sum_{t=1}^{m}j_t\Sigma_t^{-1}\mu_t\right)$

$$
= |2\pi\Sigma_i|^{-\frac{1}{2}}\prod_{t=1}^{m}\left|\frac{2\pi}{j_t}\Sigma_t\right|^{-\frac{1}{2}}\exp\left\{-\frac{1}{2}\left(x^T\Sigma^{-1}x - 2x^T\Sigma^{-1}\mu + \mu_i^T\Sigma_i^{-1}\mu_i + \sum_{t=1}^{m}\mu_t^T j_t\Sigma_t^{-1}\mu_t\right)\right\}
$$

$$
= |2\pi\Sigma_i|^{-\frac{1}{2}}\prod_{t=1}^{m}\left|\frac{2\pi}{j_t}\Sigma_t\right|^{-\frac{1}{2}}\exp\left\{-\frac{1}{2}\left(x^T\Sigma^{-1}x - 2x^T\Sigma^{-1}\mu + \mu^T\Sigma\mu - \mu^T\Sigma\mu + \mu_i^T\Sigma_i^{-1}\mu_i + \sum_{t=1}^{m}\mu_t^T j_t\Sigma_t^{-1}\mu_t\right)\right\}
$$

$$
= |2\pi\Sigma_i|^{-\frac{1}{2}}\prod_{t=1}^{m}\left|\frac{2\pi}{j_t}\Sigma_t\right|^{-\frac{1}{2}}\frac{|2\pi\Sigma|^{1/2}}{|2\pi\Sigma|^{1/2}}\exp\left\{-\frac{1}{2}\left((x-\mu)^T\Sigma^{-1}(x-\mu)\right)\right\}\exp\left\{-\frac{1}{2}\left(-\mu^T\Sigma\mu + \mu_i^T\Sigma_i^{-1}\mu_i + \sum_{t=1}^{m}\mu_t^T j_t\Sigma_t^{-1}\mu_t\right)\right\}
$$

$$
= |2\pi\Sigma_i|^{-\frac{1}{2}}\exp\left\{-\frac{1}{2}\left(\mu_i^T\Sigma_i^{-1}\mu_i\right)\right\}\prod_{t=1}^{m}\left|\frac{2\pi}{j_t}\Sigma_t\right|^{-\frac{1}{2}}\exp\left\{-\frac{1}{2}\left(\mu_t^T j_t\Sigma_t^{-1}\mu_t\right)\right\}|2\pi\Sigma|^{1/2}\exp\left\{-\frac{1}{2}\left(-\mu^T\Sigma\mu\right)\right\}\mathcal{N}(x|\mu,\Sigma)
$$

$$
= \mathcal{N}(0|\mu_i\Sigma_i)\prod_{t=1}^{m}\mathcal{N}\left(0|\mu_t, \frac{1}{j_t}\Sigma_t\right)\frac{\mathcal{N}(x|\mu,\Sigma)}{\mathcal{N}(0|\mu,\Sigma)}
$$

$\qquad\square$

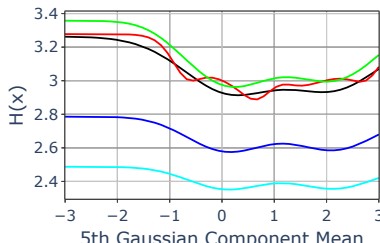

Figure 6: **Well-Behaved GMM** example is plotted on the left. The entropy of a five-component, bivariate GMM is plotted as a function of the location of the fifth component $\mu_5 \in [-3, 3]$. We show the true entropy, the $3^{\text{rd}}$ order of Huber et al.'s approximation and our two methods, along with our approximate limit.

### A.4 Extensions to Cross-Entropy

Our main results on GMM entropy approximation also extend to the cross-entropy between different GMMs. We can instead consider $H_p(q(x)) = -\mathbb{E}_{p(x)}[\log(q(x))]$ where $p(x) = \sum_{i=1}^{\widetilde{M}} \widetilde{w}_i \mathcal{N}(x|\widetilde{\mu}_i, \widetilde{\Sigma}_i)$ and $q(x) = \sum_{i=1}^{\widehat{M}} \widehat{w}_i \mathcal{N}(x|\widehat{\mu}_i, \widehat{\Sigma}_i)$. The series representations of $\log(\cdot)$ stay unchanged however we must choose a center point that allows the series to converge with respect to the inner GMM, $q(x)$. This means Theorem 3.2, Theorem 3.3, Theorem 4.2, Theorem 4.3, and Theorem 4.5 need to be reformulated so that $a > \max(q(x))$ (or the respective $a > \frac{1}{2}\max(q(x))$). We do not formally restate these theorems here for brevity. We instead provide a sketch of how the results extend to the cross-entropy setting. Choosing the bounding $a$ of the max found in Theorem 4.4 can simply be altered so that

$$\max(q(x)) \leq a = \sum_{i}^{\widehat{M}} \widehat{w}_i \left|2\pi\widehat{\Sigma}_i\right|^{-\frac{1}{2}}$$

The analogous proofs will all hold in this case. The final alteration that needs to be made is to Theorem 4.1. Again, following the exact same proof, just switching notation and being careful, one can find that

$$\mathbb{E}_p[q(x)^k] = \sum_{j_1+\cdots+j_{\widehat{M}}=k} \binom{k}{j_1,\ldots,j_{\widehat{M}}} \sum_{i=1}^{\widetilde{M}} \widetilde{w}_i \left( \frac{\mathcal{N}(0|\widetilde{\mu}_i, \widetilde{\Sigma}_i)}{\mathcal{N}(0|\mu, \Sigma)} \prod_{t=1}^{\widehat{M}} (\widehat{w}_t \mathcal{N}(0|\widehat{\mu}_t, \widehat{\Sigma}_t)^{j_t}) \right) \quad (36)$$

where $\Sigma = (\widetilde{\Sigma}_i^{-1} + \sum_{t=1}^{\widehat{M}} j_t \widehat{\Sigma}_t^{-1})^{-1}$ and $\mu = \Sigma(\widetilde{\Sigma}_i^{-1} \widetilde{\mu}_i + \sum_{t=1}^{\widehat{M}} j_t \widehat{\Sigma}_t^{-1} \widehat{\mu}_t)$. The result is the same form, however has much more convoluted notation and hence dropped from the main paper as an attempt to bring clarity to the methods being discuss without unnecessary notation.

### A.5 Recreation of Huber et al Experiment

Here we reproduce the experiment of [15] of a five-component bivariate GMM with uniform weights $w_i = 0.2$ for $i = 1,\ldots,5$, $\mu_1 = [0,0]^T$, $\mu_2 = [3,2]^T$, $\mu_3 = [1,=0.5]^T$ $\mu_4 = [2.5,1.5]^T$, $\mu_5 = c \cdot [1,1]^T$, $\Sigma_1 = \text{diag}(0.16,1)$, $\Sigma_2 = \text{diag}(1,0.16)$, and $\Sigma_3 = \Sigma_4 = \Sigma_5 = \text{diag}(0.5,0.5)$. We vary the position of the fifth mean ($\mu_5$) in the range $[-3,3]$. Fig. 6 (left) reports the third order Taylor approximations from both Taylor approximations, the Legendre approximation, as well as our approximate limiting method.

Huber et al. is accurate in the well-behaved case, but does not have any convergence guarantees nor is it a bound. Our proposed Taylor approximation sacrifices some accuracy, but is always a lower bound (Theorem 4.3) and is convergent (Theorem 4.2). We also note that our naïve limit method does gain us substantial accuracy and is still a lower bound–though we have not proven the bound property for this approximation. We notice that our Legendre approximation has comparable accuracy to that of Huber et al. in this well behaved case but has the advantage that it is guaranteed to converge (Theorem 4.5) and that we can compute higher order approximations that are difficult to define for the Huber et al. approximation.

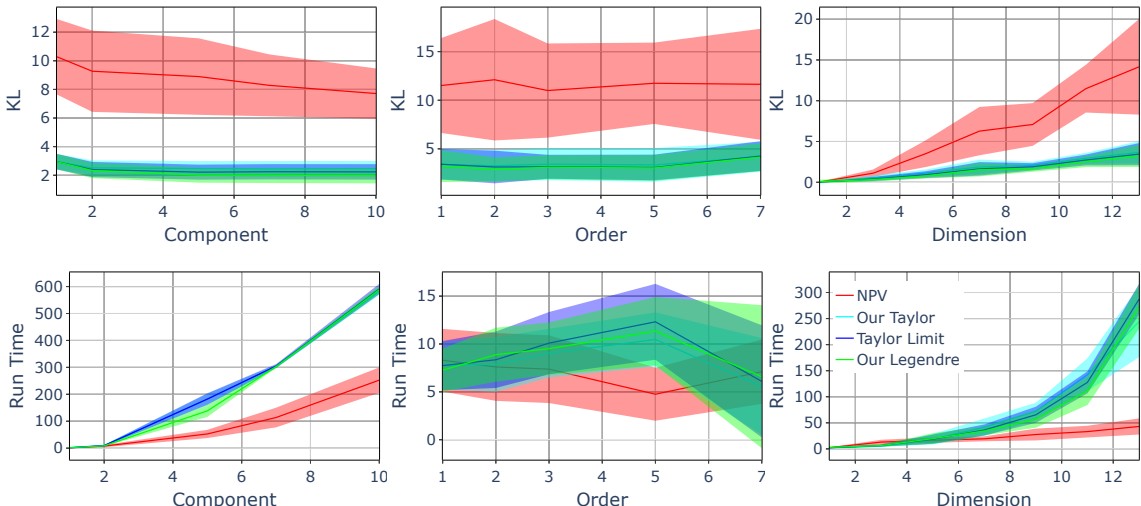

Figure 7: The above figures show the accuracy and computation time of each method across varying components, orders, and dimensions in approximating a multivariate mixture T distribution with a GMM. Our method consistently improves accuracy significantly. Higher order approximations do not increase computation time (bottom middle), and even low order approximations provide substantial improvement (top middle). Most accuracy improvements are achieved with a small number of component, unlike NPV (top left) which continues to need higher number of components to see good accuracy return. Our computation time increases with dimension due to less salable cross-entropy approximation with Gauss-Hermite quadrature (bottom right) however the guaranteed convergence of the approximation seems to have a drastic improvement on accuracy (top right).

## A.6 Nonparametric Variational Inference

Consider a target density $p(x, \mathcal{D})$ with latent variables $x$ and observations $\mathcal{D}$. The NPV approach [11] optimizes the evidence lower bound (ELBO),

$$\log p(x) \geq \max_q H_q(p(x, \mathcal{D})) - H_q(q(x)) \equiv \mathcal{L}(q) \tag{37}$$

w.r.t. a $m$-component GMM variational distribution $q(x) = \frac{1}{N} \sum_{i=1}^m \mathcal{N}(x|\mu_i, \sigma_i^2 I_d)$. The GMM entropy lacks a closed-form so NPV applies Jensen's lower bound as an approximation,

$$H_q(q(x)) = -\mathbb{E}_q \left[ \log(q(x)) \right] \geq -\sum_{i=1}^M w_i \log \left( \mathbb{E}_{q_i} \left[ q(x) \right] \right) = \hat{H}_q^J(q(x)) \tag{38}$$

The cross entropy also lacks a closed-form, so NPV approximates this term using the analogous Huber et al. Taylor approximation. Specifically, NPV expands the log density around the means of each GMM component as,

$$H_q(p(x)) \approx -\sum_{i=1}^M w_i \sum_{n=0}^N \frac{\nabla^2 \log(p(\mu_i))}{n!} \mathbb{E}_{q_i} \left[ (x - \mu_i)^n \right] = \hat{H}_{N,q}^H(p(x)) \tag{39}$$

However, Eqn. (20) is subject to the divergence criterion of Theorem 3.1 if $2p(\mu_i) \leq \max(p(x))$. This approximation is often known as the multivariate delta method for moments. The authors use these approximations of the entropy and cross entropy to create the following two approximation of the ELBO.

$$\mathcal{L}_1(q) = \hat{H}_{1,q}^H(p(x)) - \hat{H}_q^J(q(x)) \qquad \mathcal{L}_2(q) = \hat{H}_{2,q}^H(p(x)) - \hat{H}_q^J(q(x)) \tag{40}$$

Gershman et al. (2012) use the two approximations because optimizing $\hat{H}_{2,q}^H(p(x))$ with respect to the mean components, $\mu_i$ requires computing a third order, multivariate derivative of $\log(p(x))$ which is computationally expensive. The authors iterate between optimizing the mean components, $\mu_i$, using $\mathcal{L}_1(q)$ and optimizing the variance components, $\sigma_i^2$, using $\mathcal{L}_2(q)$ until the second approximate appropriately converges $\delta \mathcal{L}_2(q) < .0001$.

**In our approach**, we will highlight and address three problems with the NPV algorithm; the potential divergence of $\hat{H}_{N,q}^H(p(x))$, the inconsistent ELBO approximations, and the poor estimation of the GMM entropy via $\hat{H}_q^J(q(x))$. To address the potential divergence of $\hat{H}_{N,q}^H(p(x))$, we will take motivation from the results found in [23] and use a 2 point Gauss-Hermite quadrature method to approximate $H_q(p(x))$. This method will be a limiting factor in scaling the NPV algorithm in dimension, however it guarantees that the cross-entropy approximation will not diverge. This alteration leads to a solution for the inconsistency of the ELBO approximations. Since the quadrature method does not require any derivatives of $\log(p(x))$ w.r.t. the mean components of the GMM, we can now optimize the means and variances simultaneously on the same ELBO approximation. Finally, Jensen's inequality is a very poor approximation for entropy in general, instead we will use the three methods we have introduced, Taylor, Taylor limit, and Legendre, as the GMM entropy approximations for higher accuracy. Fig. 4 shows an approximation of a two dimensional, three component mixture Student T distribution using a fiver component GMM in the traditional NPV, using our Taylor approximation and our Legendre approximation.

**The results**, as seen in Fig. 5, highlight the accuracy and computation time of each method versus the number of components used, the order of our polynomial approximation used, and the dimension of the GMM. The accuracy is the same as seen in Section 6, the new information here is the computation time of each method. We see that the order of the method has very little impact on the computation time of our algorithm. We even see most of the accuracy improvement at around order 2 or 3 so staying in a low order approximation seems advisable. We do see that the component does increase our time by a bit compared to that of traditional NPV. The source of the computation time increase in our methods comes from more iterations in the optimization. Each evaluation of the (ELBO) approximator are near equivalent but since we are converging to a better optimum, this take more iteration steps than NPV. Finally, we see that dimension does have a large impact on our method. The source of this computation increase come from our Gauss-Hermite approximation of the cross entropy. The number of quadrature points used 2 per dimension $D$, so we are computing with $2^D$ points, which clearly scale poorly with dimension. We are seeking better ways of computing the cross-entropy of a GMM with any non-GMM function that is both convergent and computationally fast, however this was not the focus of the paper and no method was considered yet.

