# OpenReview forum: "On Convergence of Polynomial Approximations to the Gaussian Mixture Entropy"
_NeurIPS.cc/2023/Conference — NeurIPS 2023 poster_

### Official Review · Reviewer_HKUU · 2023-06-27

**Soundness:** 4 excellent
**Presentation:** 4 excellent
**Contribution:** 4 excellent
**Rating:** 7
**Confidence:** 3

**Summary:**

This paper presents new methods for approximating the entropy of a mixture of Gaussians/cross-entropy between mixtures of Gaussians. By using deterministic approximations based on power series/orthogonal polynomials expansions, the authors are able to obtain similar results as MCMC approximations for only a fraction of the computing time. Furthermore another deterministic approach (Huber) is shown to be theoretically and empirically divergent under a simple condition. The result is applied to Nonparametric Variational Inference and is shown to exhibit a better convergence behavior than the Huber approximation.

**Strengths:**

The paper is well written and the experiments are clearly presented. The theoretical claims are solid and provide useful insights and methods for approximationg entropy of GMMs. The convergence criterions clearly show the limitations of the already used Huber method and is supported empirically. Moreover the method of Taylor approximation and Legendre polynomials are shown to perform well in practice and outperform Huber is some settings.

**Weaknesses:**

- I think it could be beneficial to have a presentation of NPV from the start for the sake of motivation
- no convergence rates/error estimate is presented or at least discussed, I think this is also crucial even though harder to investigate

**Questions:**

-line 183 : what do you mean by convergence rate of \beta \alpha^n + \eta ? (edit : it is introduced in the appendix but is quite obscure in the main at the first read)
-I can't find the result on pointwise convergence for Legendre series when the function is continuous and L^2 ; can you point more directly in the book/to another reference ?

**Limitations:**

The authors have correctly adressed the main limitations of their results, namely the limitations to GMM and the problem of scaling to high orders of approximations as well as the number of components. Moreover, they do not indicate that their method should always be preferred to Huber's method but point out contexts where it could be the case.

---

> ### Author Rebuttal · Authors · 2023-08-09
>
> **line 183 : what do you mean by convergence rate of $\beta \alpha^n + \eta$ ?**
>
> From Taylor's theorem, a function, $f$, can be represented as $f=T_n+R_n$ where $T_n$ is the $n^{th}$ order Taylor series and $R_n=\mathcal{O}(\alpha^n)$ is the remainder. Rewriting this function in terms of the Taylor series, $T_n=-R_n+f= \beta\alpha^n+\eta$, we assume $\beta<0$ to force the negative on the remainder and use $0<\alpha <1$ to model the decay of the error. We agree the main text could benefit from a clearer description of this and will move the discussion in section A.5 to the main body.
>
> **I can't find the result on pointwise convergence for Legendre series when the function is continuous and $L^2$**
>
> The reference we cite in the paper is Bharwy et al. [4]. They define the shifted Legendre polynomials and state (without proof) that they are an orthonormal basis on $L^2((0,L))$. For a rigorous proof of this statement, one can look at "Linear Operators in Hilbert Spaces" by Joachim Weidmann in Section 3.2. We can add a reference to this book if the reviewer believes this should be included.
>
> **no convergence rates/error estimate is presented or at least discussed, I think this is also crucial even though harder to investigate**
>
> We agree that having some statement of convergence rate would be a nice addition, but the reviewer is correct that this is harder to investigate. We attempted to approach the Taylor series using the Taylor theorem which says that the $n^{th}$ order Taylor series has a remainder $R_{n+1}(x)= \dfrac{f^{(n+1)}(c)}{(n+1)!}(x-a)^k$ for some $c\in[a,x]$. Since we are expanding the Taylor series inside the expected value of the entropy term, we must compute $\mathbb{E}_{q}\left[\dfrac{f^{(n+1)}(c)}{(n+1)!}(x-a)^k\right]$ however $c$ is dependent on $x$. Bounding this quantity is not fruitful because as $x\rightarrow 0$, the bound diverges. Furthermore, for Legendre and Taylor, we attempted other traditional approaches to finding rates of convergence however fell into similar issues. This is still ongoing research and we hope to make this contribution in future work.

---

> > ### Comment · Reviewer_HKUU · 2023-08-20
> >
> > Thank you for your answers and I apologize for my late response. Regarding the line 183 I also think adding A.5 to the main body will be beneficial. As for the problem of approximation by Legendre serie ; this is precisely the point of my remark : the convergence is in L^2 and not pointwise ; hence a formula such as "log (p(x)) = sum... for all x" is far from obvious (for instance in the Fourier classical theory not every continuous function on (0,1) is the pointwise sum of its Fourier serie). I assume that for mixtures of Gaussians the result has good chances to be true but this requires some precisions. I would recommend stating those results almost everywhere instead. Overall my score remains unchanged.

---

> > > ### Author Response · Authors · 2023-08-21
> > >
> > > Based on the reviewer's latest clarification we agree that the statement "log (p(x)) = sum... for all x" requires further details to establish pointwise convergence more clearly.  Thank you for pointing this out.  Pointwise convergence is not necessary for our result as we utilize this approximation in Thm. 4.5 (just below L464 in the appendix):
> > > $$ \dots = -\sum_{i=1}^M w_i \int q_i(x)\log(p(x))dx = -\sum_{i=0}^M w_i \int q_i(x) \sum_{n=0}^\infty \dots L_{[0,a],n}(p(x))dx = \dots$$
> > > We will instead revise the statement that the series is convergent almost everywhere.  This result is shown in ["The Convergence Almost Everywhere of Legendre Series"](https://www.jstor.org/stable/2037625?seq=1)(Harry Pollard), which proves that "log (p(x)) = sum..." holds for x a.e. and furthermore a.e. is a sufficient condition for our proof.
> > >
> > > The series for log(p(x)) is in fact pointwise convergent everywhere except p(x)=0.  For p(x) a GMM it is everywhere positive in the domain so p(x)>0. Pointwise convergence can be shown with standard results, for example Theorem 12 of [Differential Equations and Linear Algebra](https://www.math.utah.edu/~gustafso/s2013/3150/slides/seriesMethodsLinearDE.pdf) (Gustafson, Grant B.).  Given the limited time left in the discussion period, however, we propose to change the statement to a.e. convergence as pointwise convergence is not necessary.

---

### Official Review · Reviewer_2uqK · 2023-07-05

**Soundness:** 3 good
**Presentation:** 3 good
**Contribution:** 2 fair
**Rating:** 6
**Confidence:** 3

**Summary:**

The paper provides an improved approximation to the entropy of Gaussian Mixture Models (GMMs). The proposed approximation is a lower bound to the entropy and is more accurate compared to related methods.

**Strengths:**

1. Accurate computation of the GMM entropy has implications on variational inference (VI) as well as information theory. The proposed approximation establishes a lower bound for the GMM entropy which can be naturally adapted in VI.
2. The authors show that the proposed approximation overcomes the divergence issue in the related methods, e.g., Huber et al. (2008), and hence provides stronger theoretical guarantees.
3. The method utilizes an interesting idea leveraging the Legendre series approximation of the logarithm.

**Weaknesses:**

1. While the proposed approximation establishes a lower bound for the GMM entropy, it is not clear whether the bound is tight. Thus, applying the approximation to VI may not yield monotonically increasing ELBO.
2. Computation of the approximation seems very intensive, e.g., Eq. 9. and Eq. 12. The increased accuracy may not justify the computational costs.
3. The authors only provided simulation results on toy data. It could be helpful to demonstrate the benefits of the approximation in a real-life problem.

**Questions:**

1. How do you choose the order of the approximating polynomials?
2. How does the approximation improve the VI compared to using Huber et al.?

---

> ### Author Rebuttal · Authors · 2023-08-09
>
> **While the proposed approximation establishes a lower bound for the GMM entropy, it is not clear whether the bound is tight. Thus, applying the approximation to VI may not yield monotonically increasing ELBO**
>
> The Taylor series bound on GMM entropy becomes tight as the order of the approximation increases. This is a result of convergence of the series (Theorem 4.2) and the lower bound property (Theorem 4.3).
>
> **Computation of the approximation seems very intensive, e.g., Eq. 9. and Eq. 12. The increased accuracy may not justify the computational costs.**
>
> Computation is dominated by the combinatorial sum in Eq. (11) in Lemma 4.1. For an $M$-component GMM our $n$-th order Taylor series this is dominated by $\mathcal{O}((n+M-1)!)$, see Appendix A.5 for discussion.  Computation of the series in Eq. (11) can easily be parallelized to utilize GPU acceleration, offsetting the cost.  In our experiments we achieve good accuracy with moderate polynomial orders, making GPU parallelization unnecessary.  Empirical analysis of computation time is presented in Fig. 3.
>
> **The authors only provided simulation results on toy data. It could be helpful to demonstrate the benefits of the approximation in a real-life problem.**
>
> We recognize the desire for an application of our methods to real data and this will be a focus in future work. The focus of the present work is a convergent alternative to that of Huber et al.  To support our claims made in this paper we prefer controlled synthetic evaluation, as it allows us to demonstrate divergence of the baseline, convergence of our approaches in the same setting, and straightforward evaluation of computation time.  Evaluation in real-data scenarios becomes less straightforward as the data distribution is unknown and cannot be instrumented.
>
> **How do you choose the order of the approximating polynomials?**
>
> Our series approximations are convergent, meaning higher-order polynomials are more accurate. The user may choose the order based on computational resources and required accuracy for the problem at hand. This is not necessarily true for Huber et al. as the series may diverge and divergence is more severe at higher orders. In A.6, we reproduce the toy model from Huber et al. in which that method is well-behaved.  Figure 6 shows that our Legendre series yields comparable accuracy and Figure 3 (bottom right) shows the two methods have comparable computation time.
>
> **How does the approximation improve the VI compared to using Huber et al.?**
>
> The accuracy of the Legendre polynomial is close to that of Huber et al. when the latter method is convergent. In this setting the benefit of Legendre over Huber et al. is limited, but is guaranteed to be convergent in all cases.  Our Taylor series benefits from being a lower bound on $H_q(q(x))$. In ELBO,
>
> $\log(p(x))\geq L(\theta) = \mathbb{E}\_{q_\theta(y|x)}[\log p(x,y)] + H_{q_\theta(y|x)}(q_\theta(y|x))$
>
> this lower bound on the entropy term gives a lower bound on $L(\theta)$, and thus a lower bound on $\log(p(x))$. This bound does not hold in Huber et al.'s approximation.

---

> > ### Comment · Reviewer_2uqK · 2023-08-18
> > **After rebuttal**
> >
> > Thank you for addressing my comments. I've increased my score accordingly.

---

### Official Review · Reviewer_gUEj · 2023-07-05

**Soundness:** 4 excellent
**Presentation:** 4 excellent
**Contribution:** 3 good
**Rating:** 7
**Confidence:** 4

**Summary:**

This paper looks at the convergence of an existing approximation of the GMM entropy. They show conditions, which commonly occur, in which the approximation diverges. The authors then present several new approximations in which conditions can be chosen that guarantee convergence. The authors demonstrate this both theoretically and empirically.

**Strengths:**

I have reviewed a previous version of this paper and this version is stronger in all regards. The theory is sound, the experiments are well-done, and the paper is clearly written. My concerns for previous versions of this paper have been addressed.

**Weaknesses:**

The computational complexity is a concern for this method. Also, the contribution here is somewhat limited. Other methods involving entropy often consider nonparametric estimation methods that can be adapted to different and unknown densities. Here, it is assumed that the parameters of the GMM are known. That said, the GMM is of sufficient importance in machine learning that I believe the contribution is sufficient.

**Questions:**

No questions.

**Limitations:**

The authors address the limitations.

---

> ### Author Rebuttal · Authors · 2023-08-09
>
> **I have reviewed a previous version of this paper**
>
> We sincerely thank the reviewer for their effort and helpful insights on an earlier submission.  The early feedback has strengthened the present work.
>
> **The computational complexity is a concern for this method**
>
> For similar polynomial orders of our approximations to Huber et al., the computation complexity is comparable (Figure 3). The computation cost may start to become a larger focus for higher polynomial orders due to the combinatorial sum in computing Eq. (11) in Lemma 4.1. However, the multivariate normal distribution terms can be evaluated outside the summation once as their values are constant, meaning the whole summation is simply scalar terms. This computation can then easily be parallelized to utilize GPU acceleration to offset cost.  In our experiments we found that GPU parallelization was not necessary as adequate accuracy is achieved with moderate polynomial orders.
>
> **The contribution here is somewhat limited. Other methods involving entropy often consider nonparametric estimation methods that can be adapted to different and unknown densities. Here, it is assumed that the parameters of the GMM are known**
>
> Our work builds upon the well-established approximation introduced by Huber et al., which has over 300 citations indicating its widespread recognition and use. We highlight a crucial observation that the original approach lacks general convergence. In response, we present two novel methodologies that not only ensure convergence but also extend applicability to higher polynomial orders compared to previous approaches. Furthermore, our advancements come with no additional computational overhead for similar accuracy. We believe that this firmly establishes our contribution's broader impact and its potential benefits to a wider research audience.

---

> > ### Comment · Reviewer_gUEj · 2023-08-17
> >
> > I have read the other reviews and the authors' responses. I am satisfied with their responses and am keeping my score the same.

---

### Official Review · Reviewer_BrW6 · 2023-07-22

**Soundness:** 3 good
**Presentation:** 2 fair
**Contribution:** 3 good
**Rating:** 4
**Confidence:** 5

**Summary:**

The authors show that a common entropy approximator, based on Taylor series approximators, for Gaussian mixture models is not necessarily convergent.  They propose an alternative approach based on the Taylor expansion of the logarithm, whose efficiency hinges on the fact that moments of Gaussians can be computed efficiently.

**Strengths:**

The proposed approach fills gaps in the literature and proposes novel solutions.

**Weaknesses:**

It isn't clear how practical this approach is.  Many of the lemmas and theorems are known results, and the authors do not do a good job differentiating their contributions.  The presentation could use some improvement.  The experiments are entirely synthetic despite the possibility of applying this approximation as part of non-parametric variational inference.  Computational complexity is not clearly stated.

**Questions:**

-  Practical computation time is, of course, important.  However, I think you should also include a formal statement of the computational complexity of all of the methods involved.

-  How does this approach perform when applying NPVI to a real data set?  The experiments are mostly synthetic -- it would have been nice to see a real application.

Minor typos:
While the submission is comprehensible, the language, e.g., in the introduction and elsewhere, feels a bit unnatural. I suggest a proofreading pass to smooth it out a bit.

-  "i.e." and "e.g." must always be followed by commas.
-  The sentence in lines 73-75 is awkward.
-  The sentence in line 86-87 is not a sentence
- "h" is not defined in (5).
- The presentation is a bit confusing as it seems to present the univariate Taylor theorem as applying to multivariate functions...  This is fine in (6), which is a univariate Taylor series, but the GMM problem is presented in the general multivariate case.
-  Notation changed from c to a in (6).  Maybe be consistent?
-  Line 152 "we can applying".

**Limitations:**

Yes.

---

> ### Author Rebuttal · Authors · 2023-08-09
>
> **"Many of the lemmas and theorems are known results..."**
>
> Key results of this paper are divergence of the Huber et al. approximation (Thm. 3.1) along with our Taylor and Legendre approximations and their convergence (Thm. 4.2 & 4.5).  To the best of our knowledge these results have not previously been established.  We have included some established results restated in our own context for completeness, such as convergence of Taylor (Lemma 3.2) and Legendre (Lemma 3.3) series.  We do not claim these as novel contributions of the paper and provide references where appropriate.  We can defer these results to the appendix if the reviewer feels it would improve the presentation.  We are also happy to include any references that the reviewer feels may have been overlooked if they are provided.
>
> **Practicality of approach**
>
> Our work builds on an established approximation proposed in Huber et al., which has been accepted as practical with over 300 citations at the time of this writing.  We show that this widely cited method does not converge in general and we provide two alternatives that are both convergent and applicable to higher polynomial orders than previous work.  Moreover, our approach has no additional computational overhead compared to existing work (see response below and paper Fig. 3).  We feel this firmly establishes that the work is, both, practical and beneficial to the community.
>
> **"...I think you should also include a formal statement of the computational complexity..."**
>
> Appendix A.5 outlines complexity of the Taylor approximation and motivates our proposed limit approximation.  We summarize here for convenience: Complexity of the approximation is dominated by the summation over additive sequences with a fixed sum in Eqn. (11).  For an $M$-component GMM our $n$-th order Taylor series this is dominated by $\mathcal{O}((n+M-1)!)$.  We will include this statement in the main text, along with complexity of our Legendre series, which is comparable as it is dependent on the same dominating summation.
>
> Huber et al. do not provide complexity order of their approach, but we provide a summary of our calculation here for the reviewer's convenience.  Recall that the Huber et al.'s approximation about mean $\mu_i$ is given in Eq. (5) as, $\sum_{n=0}^\infty \frac{1}{n!} \log(p(\mu_i))^{(n)} (x-\mu_i)^n$.  For $D$-dimensional r.v. $x$ the term $(x-\mu_i)^n$ requires $\mathcal{O}(D^n)$ operations. The term $\log(p(\mu_i))^{(n)}$ requires Faa' di Bruno's formula, which yields a sum over all $n$-tuples of non-negative integers $(m_1,\dots, m_n)$ s.t. $1m_1+2m_2+\dots+nm_n=n$ which is combinatorial. We will include an extended form of this discussion in the final appendix.
>
> **Synthetic vs. Real-Data Experiments**
>
> We agree with the reviewer that real-data experiments would be interesting, and will be a focus of future application of this methodology.  For the present work we feel that controlled synthetic evaluation is preferable for supporting our claims in this paper.  Our synthetic evaluation supports theoretical analysis by directly showing divergence of Huber et al. even in simple settings, validates our convergence results in those same settings, and provides a computational comparison along several dimensions that can easily be controlled.
>
>
> **The presentation...seems to present the univariate Taylor theorem as applying to multivariate functions...**
>
> Huber et al. expands about the vector-valued random variable $x$.  Because of this Huber et al. cannot easily be calculated above second-order due to tensor arithmetic that arises (see computation above).  By contrast our approach expands about the density function $p(x)$, which is always a scalar function by definition. This distinction is what allows our approach to be computed to polynomial orders above 2 without tensor arithmetic.  Divergence of Huber et al. in the scalar case (Thm. 3.1) is sufficient to establish that the method is not convergent in general.
>
> **Notation changed from c to a in (6). Maybe be consistent?**
>
> This notation change is intentional and explicitly noted following Eq. (6): "Note the change of c to a as the Taylor series center...".

---

### Author Rebuttal · Authors · 2023-08-09

We thank all reviewers for their time and useful insights.  We have provided individual responses to each reviewer below.  For brevity we do not address minor typos, all of which will be corrected in a final version.

---

### Decision · Program_Chairs · 2023-09-21

**Decision:**

Accept (poster)

**Comment:**

The reviewers agree that this paper makes two valuable contributions: showing that a widely used estimator of the Gaussian mixture model entropy is inconsistent and providing alternatives that are consistent. The claims are well supported by both theory and numerical results.